# Concurrent modes of climate variability linked to spatially compounding wind and precipitation extremes in the Northern Hemisphere

Bastien François[1,*], Khalil Teber[2,*], Lou Brett[3], Richard Leeding[4], Luis Gimeno-Sotelo[5, 6, 7], Daniela I. V. Domeisen[8, 9], Laura Suarez-Gutierrez[9, 10], and Emanuele Bevacqua[11]

[1]Royal Netherlands Meteorological Institute (KNMI), Research and Development Weather and Climate (RDWK), De Bilt, The Netherlands
[2]RSC4Earth, IEF, Leipzig University
[3]Department of Civil and Environmental Engineering, University of Strathclyde, Glasgow, UK
[4]Department of Earth Sciences, Uppsala University, Uppsala, Sweden
[5]Departamento de Estatística e Investigação Operacional, Faculdade de Ciências, Universidade de Lisboa, Lisboa, Portugal
[6]CEAUL - Centro de Estatística e Aplicações, Faculdade de Ciências, Universidade de Lisboa, Lisboa, Portugal
[7]Centro de Investigación Mariña, Environmental Physics Laboratory, Universidade de Vigo, Ourense, Spain
[8]Université de Lausanne, Lausanne, Switzerland
[9]Institute for Atmospheric and Climate Science, ETH Zurich, Zurich, Switzerland
[10]Institut Pierre-Simon Laplace, CNRS, Paris, 75252, France
[11]Department of Compound Environmental Risks, Helmholtz Centre for Environmental Research – UFZ, Leipzig, Germany
[*]These authors contributed equally to this work.

**Correspondence:** Bastien François (bastien.francois@knmi.nl) and Emanuele Bevacqua (emanuele.bevacqua@ufz.de)

**Abstract.** Compound wind and precipitation (CWP) extremes often cause severe impacts on human society and ecosystems, such as damage to crops and infrastructure. Spatially compounding events with multiple regions affected by CWP extremes in the same winter can impact the global economy and reinsurance industry, however our understanding of these events is limited. While climate variability modes such as El Niño Southern Oscillation (ENSO) can influence the frequency of precipitation and wind extremes, their individual and combined effects on spatial co-occurrences of CWP extremes across the Northern Hemisphere have not been systematically examined. Here, by combining reanalysis data and climate model simulations, we investigate how two oceanic and two atmospheric variability modes – ENSO, the Atlantic Multidecadal Variability (AMV), the North Atlantic Oscillation (NAO) and the Pacific North American (PNA) – amplify the wintertime (December–February) frequency of daily CWP extremes and associated spatial co-occurrences across the Northern Hemisphere. We find many hotspot regions where concurrent variability mode anomalies significantly amplify wintertime CWP extreme event frequencies compared to single variability modes. By examining the relationships between frequencies of wintertime CWP extremes across regions, we identify dependencies enabling extreme spatially compounding events, that is winters with many regions experiencing CWP extremes. While ENSO is the most influential variability mode for such extreme spatially compounding events, the occurrence of these events increases further when multiple modes of variability are in anomalous phases. In particular, combinations of modes increase both the number of regions and the population exposed to daily CWP extremes in the same winter. For example, combined ENSO- and NAO+ nearly doubles the number of affected regions compared to neutral con-

ditions on average. Our analysis highlights the importance of considering the interplay between variability modes to improve risk management and adapt to the impacts of spatially compounding CWP extremes.

## 1 Introduction

Weather and climate extremes pose severe risks to human health, infrastructure, natural resources, and ecosystem health (IPCC, 2021). Compound weather and climate events, defined as the combination of multiple drivers and/or hazards that contribute to societal or environmental risk, often cause more severe impacts than the respective single hazards (Zscheischler et al., 2018). The Intergovernmental Panel on Climate Change (IPCC) Special Report on Managing the Risks of Extreme Events and Disasters to Advance Climate Change Adaptation (SREX) highlighted the importance of studying compound events to improve modeling and risk estimation of weather impacts (IPCC, 2012). Without considering compound extreme weather events, damages from extreme weather events may be either under- or over-estimated (e.g., van den Hurk et al., 2023; Hillier et al., 2020). A highly studied example of an impactful compound event is hot-dry conditions (e.g., Bevacqua et al., 2022), which are enhanced by land-atmopshere feedback (e.g., Zscheischler and Seneviratne, 2017; Rasmijn et al., 2018; Ridder et al., 2022). Another example is compound wind and precipitation (CWP) extremes that can cause more damage than high winds or precipitation in isolation (e.g., Li et al., 2024). For example, the combination of high winds and rainwater can result in severe damage due to the inflow of water through joints or cracks in building structures (e.g., Blocken and Carmeliet, 2004; Mirrahimi et al., 2015; Martius et al., 2016; Jeong et al., 2020). CWP extremes can lead to a range of other exacerbated impacts, such as increased soil erosion (e.g., Foulds and Warburton, 2007), agricultural and forestry losses (e.g., Van Stan et al., 2011; Ridder et al., 2022), and damage to energy infrastructure and buildings (e.g., Mirrahimi et al., 2015; Jeong et al., 2020). They can also result in economic impacts through insurance and disaster recovery (e.g., van den Hurk et al., 2023; Ciullo et al., 2023), and human fatalities (e.g., Pilorz et al., 2023). Going forward, these CWP extremes are likely to change in frequency and/or intensity due to human-induced climate change (IPCC, 2021).

Alongside co-occurring extremes in the same location, which correspond to *multivariate events* (Zscheischler et al., 2020), another type of compound event involves the simultaneous occurrence of extremes across multiple regions or locations, referred to as *spatially compounding* extremes (Zscheischler et al., 2020). Spatially compounding events are gaining prominence due to the potential to cause widespread impacts on the global food system, disaster management resources, and (re)insurance industries by simultaneously affecting large population and asset (e.g., Singh et al., 2021; Ciullo et al., 2023; Gampe et al., 2024; Bevacqua et al., 2024). For example, widespread spatially compounding flooding throughout nine European countries in June 2013 placed high pressures on trans-national risk reduction and risk transfer mechanisms and €12 billion in losses (Jongman et al., 2014). In cases where simultaneous extreme hazards across multiple regions occur worldwide, global catastrophe risk pooling could help with reducing the severity of economic shocks (Ciullo et al., 2023). While the characteristics of spatially compounding events such as droughts (e.g., Kim et al., 2019; Singh et al., 2021) or floods (e.g., Jongman et al., 2014) have already been studied, our understanding of spatially compounding CWP extremes and their drivers is limited.

To advance our understanding of CWP extremes and their drivers, this study focuses on the Northern Hemisphere due to the high population density and the severe impacts of CWP extremes in this part of the world (e.g., Liberato, 2014; Wahl et al., 2015; Raveh-Rubin and Wernli, 2015). Such compound extremes are most frequent in coastal regions of the Pacific and Atlantic oceans (e.g., Maraun, 2016) and tend to be more frequent and intense in the winter season (e.g., Greeves et al., 2007; Hansen et al., 2019). Consequently, this study focuses on seasonal frequency of daily CWP extremes during December-February. While multiple drivers can lead to CWP extremes, their influence varies across different geographical regions (e.g., Dowdy and Catto, 2017; Zscheischler and Lehner, 2021; Li et al., 2022; Manning et al., 2024). Typical drivers of CWP extremes include atmospheric rivers (e.g., Ralph et al., 2006; Waliser and Guan, 2017), low-pressure systems (e.g., Rappaport, 2000; Seneviratne et al., 2012; Wahl et al., 2015) including tropical (e.g., Cerveny and Newman, 2000; Zhang et al., 2021) and extratropical cyclones (e.g., Raible, 2007; Owen et al., 2021) and fronts (e.g., Raveh-Rubin and Catto, 2019). Accordingly, regions prone to cyclones are particularly exposed to CWP extremes (e.g., Martius et al., 2016; Messmer and Simmonds, 2021; Owen et al., 2021). As an example of the potential costs, the extratropical cyclones Anatol, Lothar and Martin that hit Europe in December 1999 caused damages totaling 13.5 billion USD (adjusted to the value in 2012) and over 150 fatalities (Roberts et al., 2014).

At the global scale, such drivers are modulated by modes of atmospheric and oceanic variability, which represent variations in atmospheric patterns due to internal climate variability. For example, the North Atlantic Oscillation (NAO) and Pacific North American (PNA) are the leading modes of variability affecting wintertime weather in Europe (e.g., Hurrell, 1995) and North America (e.g., Wallace and Gutzler, 1981). During extreme phases of the PNA and NAO, the intensity and location of storms and moisture transport deviate from mean conditions over the Pacific-North American region (e.g., Wallace and Gutzler, 1981; Xie et al., 2020) and the Euro-Atlantic region (e.g., Hurrell et al., 2003; Lodise et al., 2022), respectively. While positive NAO phases intensify westerly winds and shift the North Atlantic storm track toward the northeast, leading to increased storm frequency and intensity over Northern Europe, negative NAO phases weaken the westerlies and amplify storm activity in the Mediterranean region (e.g., Hurrell and Deser, 2010; Priestley et al., 2023). Oceanic modes of variability such as the El Niño Southern Oscillation (ENSO) and the Atlantic Multidecadal Variability (AMV) can further modulate atmospheric modes of variability and CWP extremes. For example, ENSO can influence both the PNA and NAO (e.g., Müller and Roeckner, 2006; Jiménez-Esteve and Domeisen, 2019, 2020) as well as weather systems in many regions worldwide (e.g., van Oldenborgh et al., 2005; Zhao, 2015; Domeisen et al., 2019). The AMV, an alternation of warm and cold sea surface temperatures in the North Atlantic on decadal timescales, has been shown to influence weather in both North America and Europe (e.g., Knight et al., 2006), as well as the long-term variability of the NAO (e.g., Davini et al., 2015). There are complex teleconnections linking the ENSO, NAO, PNA and AMV modes and their influence on weather patterns across the Northern Hemisphere (e.g., Müller and Roeckner, 2006; Toniazzo and Scaife, 2006; Pinto et al., 2009; Davini et al., 2015; Trascasa-Castro et al., 2021; Hu et al., 2023; King et al., 2023; O'Reilly et al., 2024). Some of the modes are specific phenomena that are internally driven, such as ENSO, while others are a representation of the dominant weather and climate variability in a particular region, such as the NAO or the PNA.

Understanding the drivers of CWP extremes is crucial for optimising resource distribution during response efforts, yet research in this area remains limited. Several studies have explored the influence of single or combined mode of variability anomalies on precipitation and wind extremes in isolation, both at regional (e.g., Elsner et al., 2001; Abeysirigunawardena et al., 2009; Kossin et al., 2010; Grimm, 2011) and global scales (e.g., Khouakhi et al., 2016; Gao et al., 2022; Liu et al., 2024). Regarding CWP extremes, some studies have explored the influence of individual variability modes at the regional scale only (e.g., Hillier et al., 2020; Bloomfield et al., 2024), thus not allowing the assessment of the relation between concurrent climate variability modes and spatial co-occurrences of CWP extremes across several regions. In this study, we use reanalyses data and large ensemble climate model simulations from the CESM General Circulation Model (Kay et al., 2015), to investigate the effects of ENSO, NAO, PNA and AMV modes of variability, and their combinations, on the increase in the frequency of December–January–February daily CWP extremes across the Northern Hemisphere. We consider these four modes of variability due to their already-known influence on storm activity and moisture transport in the Northern Hemisphere. As ENSO operates primarily on seasonal timescales (e.g., Schmidt et al., 2001; Camberlin et al., 2001), seasonal mean indices are considered. Specifically, we (1) analyse how different modes of variability increase wintertime regional frequencies (i.e., seasonal counts) of daily CWP extremes across individual regions of the Northern Hemisphere. We then investigate spatially compounding events with many regions under high CWP extreme frequencies in the same winter by examining the effect of (2) dependencies between CWP extremes across different regions and (3) combinations of modes of variability on such spatially compounding events. Finally, we (4) inspect the atmospheric circulation anomalies associated with these compound events. To the best of the authors' knowledge, the present study is the first to investigate CWP extremes and associated spatially compounding events across the Northern Hemisphere as mediated by multiple large-scale modes of variability. We address this research gap by using large ensemble climate model simulations to provide a robust model-based analysis of the effects of rare concurrent variability mode anomalies on low probability daily CWP extremes (e.g., van der Wiel et al., 2019; Singh et al., 2021; Raymond et al., 2022; Bevacqua et al., 2023; Qian et al., 2023; Wang et al., 2023).

The study is structured as follows: Sect. 2 describes the data used, as well as the methodology to analyse the effects of variability modes on CWP extremes and spatially compounding events. Results are provided in Sect. 3. Conclusions, discussions, and perspectives for future research are presented in Sect. 4.

## 2   Data and Methods

We examine the influence of four variability modes on CWP extremes across 25 selected regions in the Northern Hemisphere defined in the SREX (Iturbide et al., 2020,  see Fig. 1). We chose these regions as they are standard reference in IPCC reports, as they encompass areas with relatively homogeneous climatic characteristics (Iturbide et al., 2020). While using these regions does not enable an explicit analysis of dependencies between local-scale CWP extremes and modes of variability, it allows for complementing IPCC assessments.

## 2.1 Model and reanalysis data

We employ model simulations from the coupled Community Earth System Model (CESM; spatial resolution of $1.25°$ by $1.25°$), which provides forty ensemble members (Kay et al., 2015). These ensemble members are derived from the CESM model under the same model physics and external forcings, with each of the members starting in 1920 and initialised from slightly different initial states, leading to different evolutions from internal climate variability (Maher et al., 2021). Considering multiple members provides a large sample size that allows for assessing the effect of internal climate variability and infrequent combinations of modes of variability on rare compound events (e.g., Bevacqua et al., 2023; Singh et al., 2021). Daily total precipitation and daily mean of wind speed are extracted for the historical period (1950–2005). Simulated data are then extended until 2019 using the emission scenario associated with a radiative forcing of $+8.5\text{W.m}^{-2}$ (RCP 8.5 scenario), resulting in a total of $70 \times 40 = 2800$ years of data. We choose daily averages rather than maxima for wind due to data availability for the CESM simulations.

To evaluate the CESM model, we employ ERA5 reanalysis data (Hersbach et al., 2020) (spatial resolution of $0.25°$) for the period 1959-2019, from which we also extract daily means of wind speed and daily total precipitation for consistency.

Seasonal indices for the two oceanic (ENSO and AMV) and two atmospheric (NAO and PNA) variability modes for both CESM and ERA5 data are calculated from monthly data using the National Center for Atmospheric Research (NCAR) data package Climate Variability Diagnostics Package (CVDP, Phillips et al., 2014). The seasonal indices for the NAO, PNA and AMV are calculated as the mean of the monthly values for December, January and February. For these indices, December is taken from the year $n-1$, while January and February are taken from year $n$, and the resulting seasonal index is assigned to year $n$. For ENSO we proceeded similarly, but used November–January averages to account for lagged effects (e.g., Kawamura et al., 2004; Li et al., 2011; Hong Lee et al., 2023). For each mode, we defined their positive and negative phases when the index is above or below its mean by $+1$ standard deviation, respectively, otherwise the phase is considered neutral.

## 2.2 Methods

### 2.2.1 CWP extremes

Many techniques have been utilized to characterize CWP extremes, with the selection of a specific method being guided by the research question. For example, the correlation between wind and precipitation has been quantified at daily to seasonal timescales (e.g., Matthews et al., 2014; Luca et al., 2017; Hillier et al., 2020; Bloomfield et al., 2023). Logistic regression models have been applied to quantify the likelihood of a precipitation extreme occurring given the presence of a wind extreme (e.g., Martius et al., 2016). Alternative approaches include examining tail dependence (e.g., Vignotto et al., 2021) and employing impact-focused metrics (e.g., Hillier et al., 2015, 2020; Bevacqua et al., 2021a). The most straightforward approaches include counting extreme wind and precipitation co-occurrences above a given percentile (e.g., Martius et al., 2016; Bevacqua et al., 2021a), or using extremal dependency measures estimating the probability of one variable being extreme given that the other one is extreme (e.g., Coles et al., 1999; Hillier et al., 2015; Owen et al., 2021).

Here, to investigate winter season (December–February) CWP extremes at the grid cell level we derived seasonal counts of daily CWP extremes, defined as wind and precipitation values simultaneously exceeding high thresholds. This results in one count per season per grid cell, which allows for investigating the effect of seasonally-averaged climate variability modes on the counts. We use the 98th percentile of wind and precipitation over the 1950–2019 period for the main analysis based on data from the CESM model. Percentile-based thresholds are frequently used to investigate climate extremes (e.g., Zhang et al., 2011; Martius et al., 2016). Following Klawa and Ulbrich (2003) and Martius et al. (2016), we chose the 98th percentile, which is a compromise to capture the most extreme events in the CESM simulations while ensuring a sufficiently large sample size for robust statistical analysis. For model evaluation, which involves both the CESM model and ERA5 reanalyses (Figs. S1-S5 of the Supplement only), we use the 95th percentiles over the 1959-2019 period – such a lower threshold allows us for a more robust evaluation. The reason for this is that, given the ERA5's limited period, extremes in the reanalysis data set are more scarce and associated statistics for very extreme events are largely affected by sampling uncertainty (Bevacqua et al., 2021b). Selecting a slightly lower threshold allows us to reduce this sampling uncertainty and thus improve confidence in assessing the model's ability to simulate extremes (e.g., Bevacqua et al., 2021b; Kelder et al., 2022; Fischer et al., 2023).

### 2.2.2 Regional and spatially compounding effects of combined variability modes

To investigate the effect of variability modes on wintertime CWP extremes, we use three different metrics. The first metric allows for assessing the effect of modes in individual regions:

- **(Metric 1)** Regionally averaged counts of CWP extremes. Wintertime CWP counts are averaged by region over land-masses, weighted by the cosine of latitude to prevent overrepresentation of grid cells closer to the poles. This count-based metric allows us to assess the effects of variability modes on individual regions separately.

As Metric 1 is derived for each region individually, the influence of variability modes on the high regional frequencies of CWP extremes across multiple regions in the same winter (i.e., spatially compounding events) cannot be deduced directly. For example, based on Metric 1, we find that the variability mode phase ENSO+ modulates regionally averaged CWP extreme frequencies for North America and Central Asia. However, a possibility could be that half of the winter seasons with ENSO+ leads to increased CWP extremes for North America only, while the other half affects Central Asia, thus not simultaneously. Examining the dependencies between counts of CWP extremes in different regions can provide preliminary information on spatially compounding events (e.g., Bevacqua et al., 2021b) because regions connected by positive dependencies tend to experience CWP extremes at the same time. Thus, as a first step for investigating spatially compounding events, we analyse dependencies in Metric 1 computed for different regions, so as to provide preliminary information on groups of regions that may be affected by CWP extremes during the same winters.

Then, to examine spatially compounding events, we use two additional metrics. We employ these metrics to investigate the effects of variability modes on regional high frequencies of CWP extremes across multiple regions in the same winter (Metric 2) and on the total population of the Northern Hemisphere exposed to CWP extremes in the same winter (Metric 3).

– **(Metric 2)** Total number of affected regions during the same winter. For a given winter, a region is considered as affected by CWP extremes when the regionally averaged count of CWP extremes (Metric 1) is above its 80th percentile derived from the distribution of the 1950-2019 period. Then, similar to Singh et al. (2021), the total number of affected regions during the same winter is counted. Although choosing a higher percentile (>80th percentile) would enable us to focus on cases where regions are more severely affected, it would considerably limit the number of regions reported as affected and, consequently, the statistical robustness of our results.

– **(Metric 3)** Total population exposure. To assess the effects of variability modes on the population of the Northern Hemisphere exposed to CWP extremes during the same winter, we calculated for each winter the weighted averages of CWP extremes $wCWP$ (hereafter referred to as "population-weighted CWP extremes") using CWP extremes and population counts at the grid cell level as follows:

$$wCWP = \frac{\sum_{i=1}^{N_{grid}} N_{CWP_i} \times Pop_i}{\sum_{i=1}^{N_{grid}} Pop_i}$$

where $N_{CWP_i}$ and $Pop_i$ are the seasonal counts of CWP events and the population count at grid cell $i$, respectively, and $N_{grid}$ is the total number of grid cells over the Northern Hemisphere. Population weighting is utilized here as a surrogate for the assets at risk that could experience damages due to CWP extremes (e.g., Bloomfield et al., 2023). For this purpose, global population counts for the year 2020 from the GPW product have been used (Columbia University, 2016).

In addition to enabling quantifying the number of affected regions depending on variability modes, Metric 2 is also used in Fig. 5b to assess the effect of the dependencies between regions on spatially compounding events. This analysis is performed by comparing the number of affected regions (i.e., Metric 2) from the original dataset with the number obtained after breaking the dependencies via randomly shuffling regional CWP extreme counts using bootstrap in all regions in time (Bevacqua et al., 2021a).

### 2.2.3   Identifying relevant effects of single and combined variability modes for the regional and spatial metrics

We quantify the effect of a positive (or negative) phase of a single variability mode of interest on wintertime CWP extremes based on the ratio between (i) CWP metrics under the positive (or negative) phase of the mode of interest, while additionally conditioning the other modes in a neutral state, and (ii) CWP metrics under all variability modes in neutral phase. Hereafter, we referred to this effect as *direct effect*. Following Singh et al. (2021), additionally conditioning all other modes in the neutral phase in (i) serves for better isolating the causal effects of the individual variability mode of interest. Specifically, such additional conditioning allows for taking into account confounding effects arising from the considered modes, still some confounding effects may remain. In particular, modes in neutral states still vary within the range of neutral conditions and we do not control for them. In addition; further effects may arise from variability mode not considered in this study. Similarly to the analysis of single variability modes, we quantify the effect of concurrent variability modes in non-neutral phases based

on the ratio between (i) CWP metrics under the concurrent modes of interest while conditioning all other modes to neutral conditions and (ii) CWP metrics under all variability modes in neutral phase. In terms of notation, we refer to concurrent climate variability modes by specifying (positive and negative) phases of NAO, PNA, ENSO and AMV in this order, with unspecified modes being in neutral states.

Although the main part of our study is based on the analysis of the metrics defined in subsection 2.2.2, we provide an overview of the direct (resp. combined) effects of modes on CWP extreme frequencies at the grid cell level in Fig. 2 (resp. Fig. 7). Results for the effects of variability modes on regionally averaged metric (Metric 1) and spatial metrics (Metrics 2 and 3) ratios are presented in Figs. 3, 4, and 6. For both regional and spatially compounding cases, we focus on the modes of variability influences leading to an enhancement in the means of the different metrics compared to neutral conditions (hereafter referred to as *positive effects*) – we focus on the increase, rather than the decrease, as this is of relevance for potential impact to society. This study does not investigate the influence of variability modes on the decrease of the metrics. When results for the different metrics are presented with box plots (Figs. 3, 5 and 6), the interquartile range and mean of the distribution are displayed.

Given the focus on four variability modes, each with three possible phases, there are eighty-one possible phase combinations, motivating the need for a synthesis of their effects. When presenting the results in Sect. 3, we focus our analysis on individual and concurrent variability modes having a *significant and positive* effect on the different metrics using permutation tests (see subsection 2.2.4 for more details on calculating the significance). To support the interpretation of the effects of concurrent variability modes on CWP metrics, in Figs. 3 and 6 we also show the effects of all univariate variability modes that contribute to the concurrent modes with significant effects regardless of whether they exert a significant effect in isolation compared to neutral conditions or not. Furthermore, to ensure robustness in the results, we disregard concurrent variability modes that occur very rarely by only considering concurrent variability modes with an empirical return period, defined as the inverse of their relative frequency, greater than 280 years (i.e., occurring for more than 10 years in our 2800 years dataset). It should be noted here that the empirical return periods for concurrent modes provided in this study are conditional as they are calculated by conditioning all other modes to neutral conditions. Therefore, such *conditional return periods*, despite providing an indication of the rarity of concurrent modes, should not be considered as absolute return periods as, by construction, they are larger than the unconditional return periods obtained without conditioning the other modes in neutral conditions. To provide an overview of the synthesis of the variability modes, effects of all possible concurrent modes are displayed for regionally averaged metric in Figs. S8-S14 (for a selection of regions for the sake of brevity) and spatial metrics in Fig. S15.

Finally, to provide insights into possible *amplified effects of concurrent variability modes* (Singh et al., 2021) relative to the effects of the modes that contribute to such combination, we identify combinations for which the average metric is higher than that of the *underlying mode sub-combinations* (note that no test here is performed, so this should be interpreted carefully, see explanations in subsection 2.2.4). For example, the concurrent modes NAO-PNA+ENSO+ are deemed to possibly have an amplified effect relative to the underlying mode sub-combinations if the average metric under NAO-PNA+ENSO+ is higher than the underlying mode sub-combinations, which are NAO-, PNA+ and ENSO+ taken in isolation, as well as the bivariate

mode sub-combinations NAO-PNA+, NAO-ENSO+ and PNA+ENSO+. The identified combinations with amplified effects are highlighted in Figs. 3, 4, 6 and 7 via dark blue box plots and boxes.

### 2.2.4 Permutation test procedure to assess significance in effects of combined variability modes

As already mentioned in subsection 2.2.3, the statistical significance of the effects of individual and concurrent variability modes on CWP metrics compared to neutral conditions is assessed using permutation tests (e.g., Bradley, 1968; Good, 2013; DelSole et al., 2017; Singh et al., 2021). Specifically, for a given CWP metric, we test whether the ratio of the average of the metric associated with a given set of phases of interest (e.g., NAO+ENSO-, set as the numerator) to the average of the metric under neutral conditions (set a denominator) is larger than one at significance level $\alpha = 0.10$ based on one-sided tests.

Compared to a lower significance level, our chosen level allows the detection of significant effects of modes of variability while reducing false negatives in the context of small sample sizes. To test the hypothesis that the average of the metric under individual and concurrent variability modes is higher than under neutral conditions, we compared the ratio obtained from the original samples with a confidence interval of the ratio obtained from data without an effect of the modes on the CWP metric. Specifically, the latter was obtained via randomly permuting without replacement the samples for both numerator and

denominator, and re-estimate the ratio from the resampled data. By repeating this procedure, we can then define a confidence interval for the ratio and a critical region for test rejection. If the original ratio is higher than the $(1 - \alpha) * 100$th percentile, the average of the metric associated with the set of phases of interest is considered to be significantly higher than that of the neutral conditions. As several tests are carried out for the different concurrent modes, a Bonferroni correction (e.g., Bonferroni, 1936; Sedgwick, 2014) is applied to control the overall probability of Type I (or false positive) errors. Please note that, while

identifying concurrent variability modes with significantly *amplified* effects relative to the effects of the modes that contribute to such combination is relevant, it has not been done in this study as no statistically consistent procedure including Bonferroni correction has been found for such a statistical problem. For example, to identify the concurrent modes NAO+ENSO+AMV+ as having a significantly amplified effect on the metrics, six permutation tests comparing the effects of NAO+ENSO+AMV+ against those of its six sub-combinations must all reject the equality of effects. Designing such a statistical test procedure while

controlling the false discovery rate goes beyond the scope of the present study and is therefore not included in the analysis. In addition, in Figs. 2 and 7 only, we assess whether averages of the CWP extreme frequencies at the grid cell level associated with a given set of phases are significantly different compared to neutral conditions (i.e., not necessarily larger). For these tests performed at the grid cell level, we use two-sided permutation tests. In this case, the critical region for rejection is defined using the $(\frac{\alpha}{2}) \times 100$th and $(1 - \frac{\alpha}{2}) \times 100$th percentiles.

Regarding the number of samples $m$ for permutation tests, several trials showed that choosing $m = 100,000$ allows us to obtain robust results for the three different metrics. Thus, the analysis of the three metrics was carried out using $m = 100,000$. However, when applied to the grid cell level for Figs. 2 and 7, results from permutation tests proved less sensitive to the choice of the number of sample, and $m$ was chosen to be equal to 100.

## 3  Results

Before investigating the effect of individual variability modes and concurrent modes on CWP extremes and associated spatially compounding events in CESM simulations for the 1950-2019 period, we carried out a model evaluation with respect to ERA5 over the 1959-2019 period. Such a model evaluation is mainly performed for the effects of individual variability modes, as a robust assessment of the effects of concurrent variability modes requires a large sample size (Singh et al., 2021). For the same reason, the effects of individual variability modes in ERA5 are evaluated without constraining the other modes to be in the neutral phase (which thus differs from the *direct effects* of variability modes defined in subsection 2.2.3).

Results for model evaluation are displayed in Figs. S1-S5 of the Supplement. Although the simulated CWP absolute frequencies exhibit some biases with respect to ERA5 over the Northern Hemisphere (Fig. S1), CESM provides an adequate representation of the anomalies in CWP frequencies induced by individual variability modes relative to neutral conditions (Figs. S2-S5). We conclude from this model evaluation that CESM is suitable for further investigating the CWP extremes and their relationships with variability modes. In the following, we assess the effect of modes of variability and their combinations on CWP extremes via the CESM model based on the 98th percentiles, and also provide a one-to-one comparison with ERA5 based on the same percentiles.

### 3.1  Direct effects of variability modes on regional CWP extremes

Examining the direct effects of variability modes on CWP extremes in CESM simulations (Fig. 2) shows that different phases of NAO, PNA and ENSO significantly modulate CWP extreme frequencies over multiple regions of the Northern Hemisphere. Within Europe, NAO+ and NAO- significantly increase CWP extremes in Northern Europe (NEU) and the Mediterranean region (MED), respectively (Figs. 2a and 2b). NAO+ also significantly influences CWP extremes in the Russian-Arctic (RAR) and Western Siberia (WSB). The low pressure associated with storms under NAO+ is accurately represented in CESM simulations (Fig. S6a), with negative pressure anomalies partially or fully covering the regions experiencing a significant increase in CWP extremes. An exception to such CWP enhancement associated with low pressure is found for Northeastern North America (NEN) during NAO-, where high sea level pressure anomalies of $\sim +3$ hPa are associated with increased CWP extremes. While no regions experience increased CWP extremes under PNA+ (Fig. 2c), PNA- significantly increases CWP extremes compared with neutral conditions in Eastern Siberia (ESB) and Russian Far East (RFE) within North Asia (Fig. 2d), and in Northeastern North America (NEN), Northwestern North America (NWN) and Western North America (WNA) in the North America macroarea.

Among the oceanic modes of variability ENSO and AMV, the former has the most relevant and widespread effects. ENSO+ (Fig. 2e) increases CWP extremes across three regions of Africa (Arabian Peninsula ARP, Sahara SAH and Western Africa WAF). For ENSO- (Fig. 2f), CWP extremes are significantly increased for two regions within Central-South Asia: South Asia (SAS) and East Asia (EAS). For the AMV, we find no significant regional effects of AMV+ and AMV- conditions (Figs. 2g and 2h), in line with weak sea level pressure anomalies associated with AMV phases, ranging from $\sim -1.5$ to $\sim +1.5$ hPa (Figs. S6g and S6h).

## 3.2 Effects of concurrent anomalies in variability modes on regional CWP extremes

Model simulations (CESM) show that not only individual variability modes can have effects on regional wintertime frequencies of CWP extremes, but also combinations of modes (Fig. 3; Fig. S7 shows the same but for absolute frequencies). Figure 4 provides a summary of these effects, including the count of modes combinations that enhance CWP extremes compared to neutral conditions for each region. For example, Northern Europe (NEU) exhibits the highest count of mode combinations that significantly enhance CWP extreme frequencies compared to neutral conditions, with eleven mode combinations that all involve NAO+. The presence of a given mode phase or combination of mode phases that has an effect on CWP extremes in multiple regions suggests the potential for spatially compounding events, which will be examined in subsection 3.3. In the following, we focus on describing the effects of a selection of mode combinations and regions in Figs. 3-5. To maintain clarity and conciseness, we do not discuss all regions and mode combinations in the text, and readers can explore specific regional effects directly in the figures.

For the Mediterranean region (MED) in Europe (Fig. 3a), five concurrent variability modes containing NAO- have significant positive effects with respect to neutral conditions. Among these five concurrent variability modes, three combinations of modes have an amplified effect relative to the underlying mode sub-combinations (blue boxes). For example, in the MED region, CWP extremes are on average $\sim 1.5$ times more likely under the concurrent mode NAO-ENSO+ than under neutral conditions, while being $\sim 1.1$ and $\sim 1.2$ times more likely than under NAO- and ENSO+ in isolation, respectively. Note that the conditional empirical return period of NAO-ENSO+ (i.e., conditioning all other modes to neutral conditions) is 65 years, while it is equal to 20 and 30 years for NAO- and ENSO+, respectively, which means that the concurrent mode NAO-ENSO+ (with other modes being in neutral conditions) is $\sim 3$ and $\sim 2$ times less likely to occur in any given year than NAO- or ENSO+ in isolation (with other modes being in neutral conditions).

Concurrent modes can also amplify CWP extreme frequencies in other regions. For example, in the Arabian Peninsula (ARP) within Africa, CWP extremes are on average $\sim 3.5$ times more likely than neutral conditions under ENSO+. However, when ENSO+ combines with NAO-PNA+, CWP extremes are on average $\sim 4$ times more likely, while both individual modes (NAO- and PNA+) do not have a significant effect on the mean frequency of CWP extreme frequencies relative to neutral conditions. Interestingly, the combination of NAO-ENSO+ has an amplified effect relative to the underlying mode sub-combinations in all three African regions, increasing approximately the likelihood of CWP extremes frequencies by a factor of $\sim 4$, 1.5 and 2 relative to neutral conditions in the Arabian Peninsula (ARP), Sahara (SAH) and Western Africa (WAF), respectively.

The Central North America (CNA) region illustrates how the effects of individual modes can combine to significantly increase CWP extreme frequencies in a given winter relative to neutral conditions, even if individual modes do not exert significant effects in isolation. For example, despite the fact that both PNA+ and ENSO+ in isolation do not significantly increase CWP extremes (indicated by no grey arrows in Fig. 3b), combined PNA+ENSO+ increases significantly the likelihood of CWP extremes by a factor of $\sim 1.5$ compared to neutral conditions (indicated by the grey arrow and blue box). Furthermore, within North America, the region with the most combinations of modes that present amplified effects on CWP extreme frequencies is

East North America (ENA), with six separate combinations (see blue boxes). Of these combinations, ENSO is always in the positive phase, indicating its potentially large effect on CWP extremes.

In North Asia, concurrent variability modes increase CWP extreme frequencies by $\sim 1.5$ to $\sim 2$ times relative to neutral conditions depending on the combination. In particular, PNA-ENSO- increases CWP extreme frequencies by $\sim 1.5$ times in three of North Asia's four regions (Eastern Siberia ESB, Russian-Arctic RAR and Russian Far East RFE).

Within Central South Asia, the Tibetan-Plateau (TIB) exemplifies again how individual modes without significant effects on CWP extremes (here, AMV- and ENSO+) can combine to significantly increase the likelihood of CWP extremes during winters. Across the five regions of Central South Asia, all concurrent modes that, on average, led to more CWP extremes than their underlying mode sub-combinations (blue boxes) include the variability mode ENSO. Of these concurrent modes, ENSO is always in the positive phase within the inland regions (ECA, TIB and WCA); while being in the negative phase in the coastal
regions (EAS and SAS).

Regarding the comparison of CESM results with those of ERA5, the range of CESM ratios of regionally averaged CWP extreme frequencies with respect to neutral conditions generally covers the values for ERA5 for the different concurrent modes, when available (Fig. 3).

Results from Fig. 3 and the summary in Fig. 4 cannot be used to conclude whether the effects of concurrent variability modes
lead to spatially compounding CWP extremes, that is, to high wintertime frequencies of CWP extreme across multiple regions during the same winter. These figures illustrate the effect of individual and concurrent variability modes on regionally averaged CWP extreme frequencies, which are derived for each region separately. Nevertheless, the number of regions where each mode combination has significant effects in Fig. 4 (see numbers on the top of the matrix) suggests that some mode combinations may potentially lead to spatially compounding CWP extremes. For example, NAO-ENSO+ significantly enhances regional CWP
extreme frequencies in eight regions, which means that if these regional effects of NAO-ENSO+ can manifest in the same winter, NAO-ENSO+ would lead to spatially compounding CWP extremes. In the next section, we assess whether individual and concurrent variability modes can lead to concurrent CWP extremes during the same winters across regions.

### 3.3  Effects of concurrent variability modes on spatially compounding CWP extremes

As the first step in the investigation of spatially compounding events, we examine dependencies between counts of CWP
extremes in different regions, which provides preliminary information on groups of regions that may be affected by CWP extremes during the same winters. Figure 5a shows Spearman correlations of regionally averaged CWP extreme frequencies (Metric 1) between all pairs of regions. Regions that are geographically close tend to be positively correlated (see correlation values in contoured black boxes), in line with spatial autocorrelation. However, also some regions that are from different macroareas and therefore more distant can be correlated, e.g., Central-South Asian regions with African regions or North
American regions with North Asian regions. This highlights the potential for underlying effects of modes of variability that can connect distant regions. Notably, Fig. 5a highlights that most of the pair correlations between CWP extreme counts in different regions are positive.

In line with such dominant positive correlations (Fig. 5a), we find that dependencies among regions overall enhance the potential for spatially compounding events. Specifically, the distribution of the number of regions affected during the same

winters (Metric 2) is different from that obtained by assuming independence between regions (Fig. 5b), with dependencies elongating both tails of the distribution. For spatially compounding extremes on the right tail of the distribution, this implies a higher 50-year return level for the number of regions affected by high CWP counts in the same winter compared to independence, that is 11 instead of 9 affected regions (see vertical lines). Consistent with findings in Bevacqua et al. (2021a), this dependency-driven shift is not observed for the mean of the distribution (not shown).

The increased count of regions under CWP extremes due to the dependencies among regions can be linked to modes of variability. That is, the high number of regions simultaneously affected by CWP extremes, which is possible due to the dependencies among regional CWP extremes, would also not be possible without the effects of concurrent variability modes (Fig. 5c). In particular, when all variability modes are neutral, on average around four regions are affected simultaneously by high regionally averaged CWP extremes. Generally, as the number of variability modes in anomalous conditions increases,

the likelihood of multiple regions simultaneously experiencing extreme conditions also increases (Fig 5c; see letters indicating significant differences among distributions, with the exception of the distributions related to three and four modes). Notably, this effect is even more marked when focusing on winter seasons with an extreme number of regions affected (right whiskers of the boxplots). Overall, these results indicate that concurrent anomalies in variability modes are key for spatially compounding events.

Given the dependencies among regions and that combinations of variability modes are essential for spatially compounding CWP extremes, we move to identify which are the relevant mode combinations that enhance the number of regions affected by CWP extremes (Metric 2) and population exposed to CWP extremes (Metric 3). We find that thirteen individual and concurrent variability modes significantly increase the number of affected regions compared to neutral conditions (Fig. 6a). Three bivariate combinations have a significant effect on the total number of affected regions and an amplified effect relative to their underlying

mode sub-combinations (see blue boxes for ENSO+AMV-, PNA+ENSO+ and NAO-ENSO+). NAO-ENSO+ nearly doubles the number of regions simultaneously exposed to CWP extremes on average relative to neutral conditions.

In terms of population exposed to CWP extremes, four combinations (ENSO-AMV+, NAO-PNA+ENSO+, NAO+ENSO-AMV+ and PNA-ENSO-) are identified as having a significant effect compared to neutral conditions (Fig. 6b). Among these four combinations, only one (NAO-PNA+ENSO+) has already been identified as having a significant effect on the number

of affected regions in Fig. 6a. It highlights that the heterogeneous distribution of population density across regions needs to be considered to assess the societal vulnerability to CWP extremes. In particular, variability modes in isolation do not lead to significant effects on the population exposure compared to neutral conditions (Fig. 6b), indicating the importance of considering combinations of modes to distil the effects of modes of variability on the population affected. Notably, while ENSO+ dominates the influence of modes of variability on the number of affected regions (see "+" sign in Fig. 6a), ENSO- dominates for the

population affected (see "−" sign in Fig. 6b).

### 3.4 Physical mechanisms underlying spatially compounding CWP extremes

We now move to inspect the physical mechanisms leading to spatial co-occurrences of CWP extremes by analysing SLP anomalies for concurrent variability modes with significant effects on spatially compounding events (Fig. 7). Here, negative SLP anomalies during winter are generally considered indicative of storminess, a key driver of CWP extremes. In the analysis, we restrict to the seven combinations having an amplified effect relative to their underlying mode sub-combinations on Metric 2 and Metric 3 (i.e. combinations with blue boxes in Fig. 6): namely, NAO-ENSO+, PNA+ENSO+ and ENSO+AMV- for affected regions (Fig. 6a) and PNA-ENSO-, ENSO-AMV+, NAO+ENSO-AMV+ and NAO-PNA+ENSO+ for exposed population (Fig. 6b). These combinations have a reasonably low conditional return period ($\leq 82$ years; see Methods), thus are not very unlikely events in the model simulations.

The largest positive effects on the number of affected regions (Fig. 6a) occurs under concurrent ENSO+ (El Niño phase) and NAO-, that is NAO-ENSO+ (Figs. 7a and 7b). Under such a combination of modes of variability NAO-ENSO+, negative SLP anomalies patterns intensify and expand over the North Atlantic Ocean compared to those for modes NAO- and ENSO+ in isolation (Figs. S6b and S6e), in line with the enhancement of CWP extremes over many regions of North America, North Central America, Europe and Africa.

Although the combination of modes PNA+ENSO+ has smaller significant positive effects on the number of affected regions than that of NAO-ENSO+ (Fig. 6a), PNA+ENSO+ may be more frequent than NAO-ENSO+ (given their lower conditional return period; see Methods). Such a PNA+ENSO+ combination (Figs. 7c and 7d) intensifies the negative SLP anomalies patterns on both sides of North America with respect to PNA+ and ENSO+ modes in isolation (Figs. S6c and S6e). This is consistent with the amplification of CWP extreme frequencies for regions within North America, Central America and Africa. In contrast, when ENSO+ combines with AMV- (ENSO+AMV-; Figs. 7e and 7f), negative SLP anomalies patterns are quite similar to those of ENSO+ in isolation (Fig. S6e); this suggests that AMV- has a limited amplifying effect on CWP extremes, in line with Fig. 6a.

Under the resulting combination PNA-ENSO- (Figs. 7g and 7h), which shows the largest positive effects on the population exposure (Fig. 6b), negative SLP anomalies are intensified over North Asian region, compared to those for modes in isolation (Figs. S6d and S6f). These intensified low-pressure conditions associated with storminess increase CWP extreme frequencies over a number of densely populated regions in North Asia and East Asia. PNA-ENSO- exemplifies how combined variability modes can lead to spatially compounding events by adding up the effects of individual modes in isolation. Specifically, under PNA-ENSO- the population exposed to extreme CWP extremes under PNA- and ENSO- in isolation partially adds up, resulting in spatially compounding events with a large population affected by CWP extremes.

Although the effects of combined variability modes ENSO-AMV+ (Figs. 7i and 7j) are rather limited, when combined with NAO+ (NAO+ENSO-AMV+; Figs. 7k and 7l) negative SLP anomalies are intensified over high latitudes, mainly due to NAO+ influence (Fig. S6a) and over South Asia, due to ENSO- (Fig. S6f). These intensified low-pressure systems lead to an amplification of CWP extremes in Northern Europe and South Asia. It is interesting to note that these two regions are not the same affected by ENSO-AMV+ (Figs. 7i-j), suggesting a strong influence of the NAO on regions impacted by CWP extremes.

Similar to what is observed for NAO-ENSO+, NAO-PNA+ENSO+ features a negative SLP anomaly extending across the Southern North Atlantic (Fig. 7n). However, for some regions, the addition of PNA+ to NAO-ENSO+ tends to weaken the effects of concurrent variability modes on CWP extremes, as some regions exhibit, on average, fewer regional CWP extremes than those obtained under NAO-ENSO+ (see blue boxes in Figs. 7n and 7m).

## 4    Discussion

Several results regarding the direct effects of variability modes on CWP extremes in CESM simulations (subsection 3.1) align with well-established findings in the literature, thereby validating the robustness of CESM modeling. Table S1 provides a concise summary of the agreement between CESM simulations and existing literature. Among these results are the influence of the NAO+ regime favoring CWP extremes in Northern Europe due to stronger than average westerly winds (e.g., Hurrell and Deser, 2010), or the influence of PNA- favoring blocking in the Pacific (e.g., Li et al., 2017), increasing storm frequency in the

Northern Pacific. However, several results lack direct support from existing literature. While they may represent novel findings, they should be interpreted cautiously, considering the biases of CESM simulations relative to ERA5. Notably, the direct effect of ENSO+ on CWP extreme frequencies in CESM simulations exhibits some inconsistencies when compared to ERA5 over Northern Africa (Fig. S3). Better understanding and confirming the influence of climate modes on arid regions (e.g., Northern Africa), where CWP events may be less intense than in other areas, can support adaptation and mitigation policies. While CWP

extremes can serve as an important source of freshwater (e.g., Berdugo et al., 2020), they also present a significant flood risk (e.g., Yin et al., 2023).

We analysed event counts aggregated over winter and at the scale of predefined SREX regions, given that high counts of compound extremes at these scales are expected to have negative effects on society. While the 98th percentile has been used in this study to focus on extremes and is relatively well-established in the literature (e.g., Klawa and Ulbrich, 2003; Martius

et al., 2016), other higher thresholds could have been chosen to consider more intense extreme events (e.g., Liu et al., 2013; Schär et al., 2016; Camuffo et al., 2020). Figs. S16-S19 show results from a sensitivity analysis on the influence of variability modes on regional CWP extremes (Metric 1) and spatially compounding events (Metrics 2 and 3) with the 99th and 99.5th percentiles used as thresholds. Although there are some variations in the results compared to those for the 98th percentile, the main conclusions drawn across the different thresholds are broadly consistent for all Metrics. The magnitude of the effects of

the combinations are generally consistent across thresholds, and the combinations detected at higher thresholds are generally included among those identified at lower thresholds (Figs. S16-S19). Such slight differences may be due to larger sampling uncertainty for higher thresholds limiting the ability to detect significant effects for higher thresholds rather than different physical mechanisms involved for different thresholds. While the sensitivity analyses broadly indicate the robustness of most of our findings, possible relevant differences across thresholds highlight the importance of identifying impact-relevant thresholds,

though this task is challenging (e.g., Williams, 1978; Bloomfield et al., 2023). In addition, the selected SREX regions may not reflect the natural spatial patterns of variation of CWP extremes, potentially occurring at a more localized scale or span across multiple regions. We also note that variability modes such as AMV or ENSO can have lagged effects on regional climate

extremes (e.g., Ruiz-Barradas et al., 2000; Wang, 2019; Xing et al., 2022), an aspect that has been partially taken into account in this study for ENSO. Furthermore, additional variability modes than those considered here (e.g., the Indian Ocean Dipole (IOD), Chongyin and Mingquan, 2001; Qiu et al., 2014; Kurniadi et al., 2021) may be related to CWP frequencies in some regions, especially in the regions where the different combinations of the variability modes NAO, ENSO, PNA and AMV had no significant effects (in Fig. 3). In addition, the IOD is known to co-vary with some of the modes examined here, e.g. with ENSO and the AMV (e.g., Ashok et al., 2003; Stuecker et al., 2017; Xue et al., 2022). Overall, although our aggregation in time and space may not be optimal for providing a fine-grained analysis of CWP events and additional modes may be relevant in some regions, this study provides a first comprehensive assessment of the interactions between multiple climate variability modes and the frequency of wintertime CWP extremes and associated spatially compounding events across regions of the Northern Hemisphere. In particular, analyzing spatially compound events highlights the potential influence of variability modes that can link distant regions. These long-range relationships modulated by mode combinations can be explored in more detail, for example, with tailored experiments such as nudged atmospheric simulations.

We mainly focused on the increase in the frequency of CWP extremes, however we note that increases in intensity at the same frequency, as well as low frequencies of CWP extremes may also be relevant information for the insurance industry (e.g., Ciullo et al., 2023). Studying the influence of variability modes on the intensity of the drivers (here, wind and precipitation) could further support mitigation and adaptation strategies (e.g., Whan and Zwiers, 2017; Li et al., 2022). Quantifying these effects on wind and precipitation in isolation would allow for a decomposition of the effects of concurrent variability modes, giving an even more complete picture of the processes involved (e.g., Manning et al., 2018; Brunner et al., 2021; Calafat et al., 2022). In addition, regarding spatial metrics, considering other relevant measures that combine return period information with the exposed population would allow for focusing on concurrent variability modes that regularly expose the population to CWP extremes.

With the methodology used here, it is not possible to thoroughly examine the causality behind the dependence between different variability modes and between the modes and CWP extremes. It is however clear that there exist important interactions and causal links between oceanic and atmospheric variability modes at different time scales, e.g., between ENSO and the NAO (e.g., Kirov and Georgieva, 2002; Deser et al., 2017; Yeh et al., 2018), between the AMV and the NAO (e.g., Hurrell and Deser, 2010; Fang et al., 2018) and between ENSO and the PNA (Renwick and Wallace, 1996). Therefore, these modes are dependent on each other, a relevant information to take into account when investigating the causal effects of climate variability modes on characteristics of spatially compounding events. Applying statistical methods such as regression techniques (e.g., Pearl, 2013; Kretschmer et al., 2021), or more advanced approaches such as causal networks (e.g., Nowack et al., 2020) may help to shed light on the complete causal pathway leading to spatially compounding events and better control potential confounding effects. We also note that dependencies among modes influence the return periods of concurrent modes, that is, certain signs of modes will co-occur with others more frequently than others, such as e.g. ENSO+NAO-, with an effect on their associated CWP extreme frequencies. In line with heterogeneous dependencies among different modes, we estimated a wide range of return periods for the different mode combinations (Fig. 3).

A natural continuation of this work is the application of the methodology developed in this study to investigate 1) the influence of climate modes on CWP extremes in the Southern Hemisphere, 2) summertime CWP extremes and 3) changes in CWP extremes over time, as well as their spatial relationships and dependencies with climate variability modes under climate change (Bevacqua et al., 2023).

## 5 Conclusions

In this study, we present the first comprehensive assessment of the relation between individual and concurrent climate variability modes and the frequency of wintertime CWP extremes and associated spatially compounding events across regions of the Northern Hemisphere using CESM simulations. We show that simulated concurrent modes are associated with an amplification of CWP extreme frequencies in many individual regions compared to variability modes in isolation. We have identified groups of regions with positive spatial correlations between regional wintertime CWP extremes. These correlations enable extreme spatially compounding events with many regions experiencing CWP extremes during the same winter. We found that such extreme spatially compounding events, which also include a large fraction of the population of the Northern Hemisphere under CWP extremes in the same winter, are possible due to the combinations of multiple variability modes in anomalous phases and their influence on the atmospheric circulation. For example, the combination of ENSO- and NAO+ nearly double the number of affected regions compared to neutral conditions on average. Among the modes, ENSO is found to be the most influential variability responsible for spatially compounding extreme events, which aligns with its known effects on weather and climate extremes worldwide (Goddard and Gershunov, 2020). The high return periods associated with some of the concurrent modes that lead to extreme spatially compounding events make it potentially difficult to factor these findings into long-term planning, e.g., infrastructure development or (re)insurance modelling. However, by identifying the drivers of the most extreme events, the findings raise awareness on the potential for extreme compounding events under certain mode combinations, which could be factored into weekly, sub-seasonal (White et al., 2022), seasonal (Lenssen et al., 2020) and longer forecasts to better anticipate mitigation action and climate services (Osman et al., 2023).

By using two different metrics to characterise spatially compounding events, we highlight that the effects of concurrent variability modes can differ from one spatial metric to another. While combinations with ENSO+ lead to the largest number of affected regions, when weighted by population exposure, combinations with ENSO- lead to higher effects on population. Our analysis thus stresses the importance of considering not only the interplay between variability modes but also a careful choice of metrics, which should be tailored to the ultimate impacts of interest, so as to assess the relevant characteristics of spatially compounding events and improve their risk management and mitigation.

*Data availability.* The ERA5 reanalysis data can be accessed via the "Climate Data Store" (CDS) web portal https://cds.climate.copernicus. eu/cdsapp#!/dataset/reanalysis-era5-single-levels. The CESM Large Ensemble Simulations can be downloaded from the CESM Large En-

semble Community Project website: https://www.earthsystemgrid.org/dataset/ucar.cgd.ccsm4.CLIVAR_LE.html. Modes of variability for CESM can be accessed via the CVDP-LE Data Repository https://www.cesm.ucar.edu/projects/cvdp-le/data-repository.

*Author contributions.* EB designed the initial plan of the study with DD. EB, LSG and DD supervised the project. EB, LSG, BF and KT
designed the experiments and the statistical analyses with inputs from all co-authors. EB, KT and LSG provided the code for data pre-processing. BF and KT made all computations and figures. BF, LB, RL, KT and LGS made the analyses and interpretations with inputs, corrections and additional writing contributions from EB, LSG and DD. BF wrote the first draft of the article with inputs from all co-authors.

*Competing interests.* The authors declare that they have no conflict of interest.

*Acknowledgements.* EB received funding from the Deutsche Forschungsgemeinschaft (DFG, German Research Foundation) via the Emmy
Noether Programme (grant ID 524780515). LSG received funding from the European Union's Horizon Europe Framework Programme under the Marie Sklodowska-Curie grant agreement No 101064940. This work by LGS is partially financed by national funds through FCT – Fundação para a Ciência e a Tecnologia under the project UIDB/00006/2020, DOI: 10.54499/UIDB/00006/2020, and project UID/00006/2025. This project has received funding from the European Union's Horizon 2020 research and innovation programme under grant agreement No 101003469. This work emerged from the Training School on Dynamical Modelling of Compound Events organized by the European
COST Action DAMOCLES (CA17109). This project has received funding from the European Research Council (ERC) under the European Union's Horizon 2020 research and innovation programme (Grant agreement No. 847456). Support from the Swiss National Science Foundation through project PP00P2_198896 to DD is gratefully acknowledged. We thank the NCAR's Climate Analysis Section for producing and making available the Climate Variability Diagnostics Package and the CESM simulations.

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

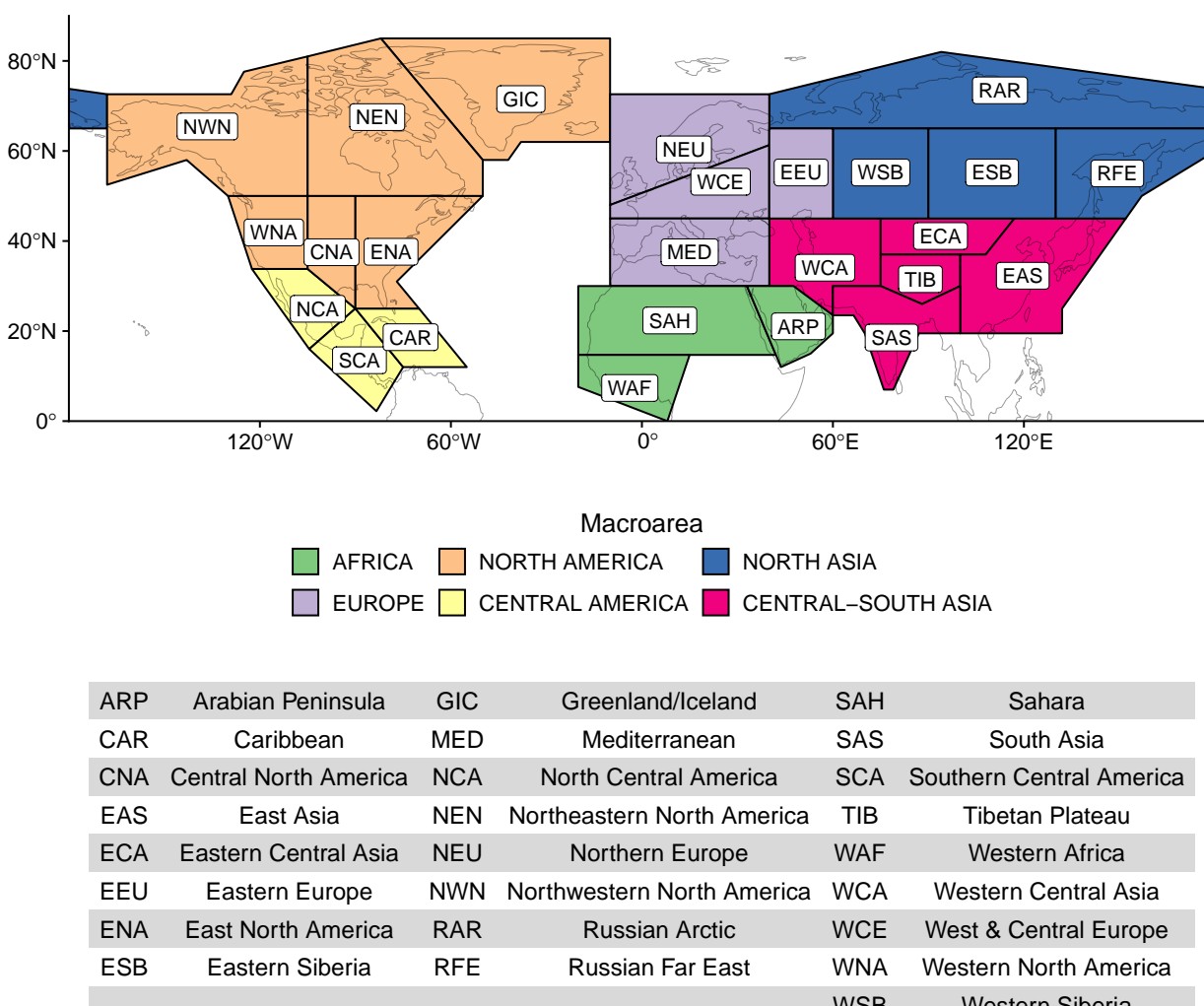

**Figure 1.** Presentation of the regions under study, that is the SREX regions (Iturbide et al., 2020) adopted by the Intergovernmental Panel on Climate Change (IPCC) in the Northern Hemisphere. The bottom part shows short and full names of the regions. Regions are clustered in macroareas (see legend). To balance the number of regions between continents, we partitioned the Asian continent into two macroareas (North Asia and Central-South Asia) and included the Arabic Peninsula (ARP) region in the Africa macroarea.

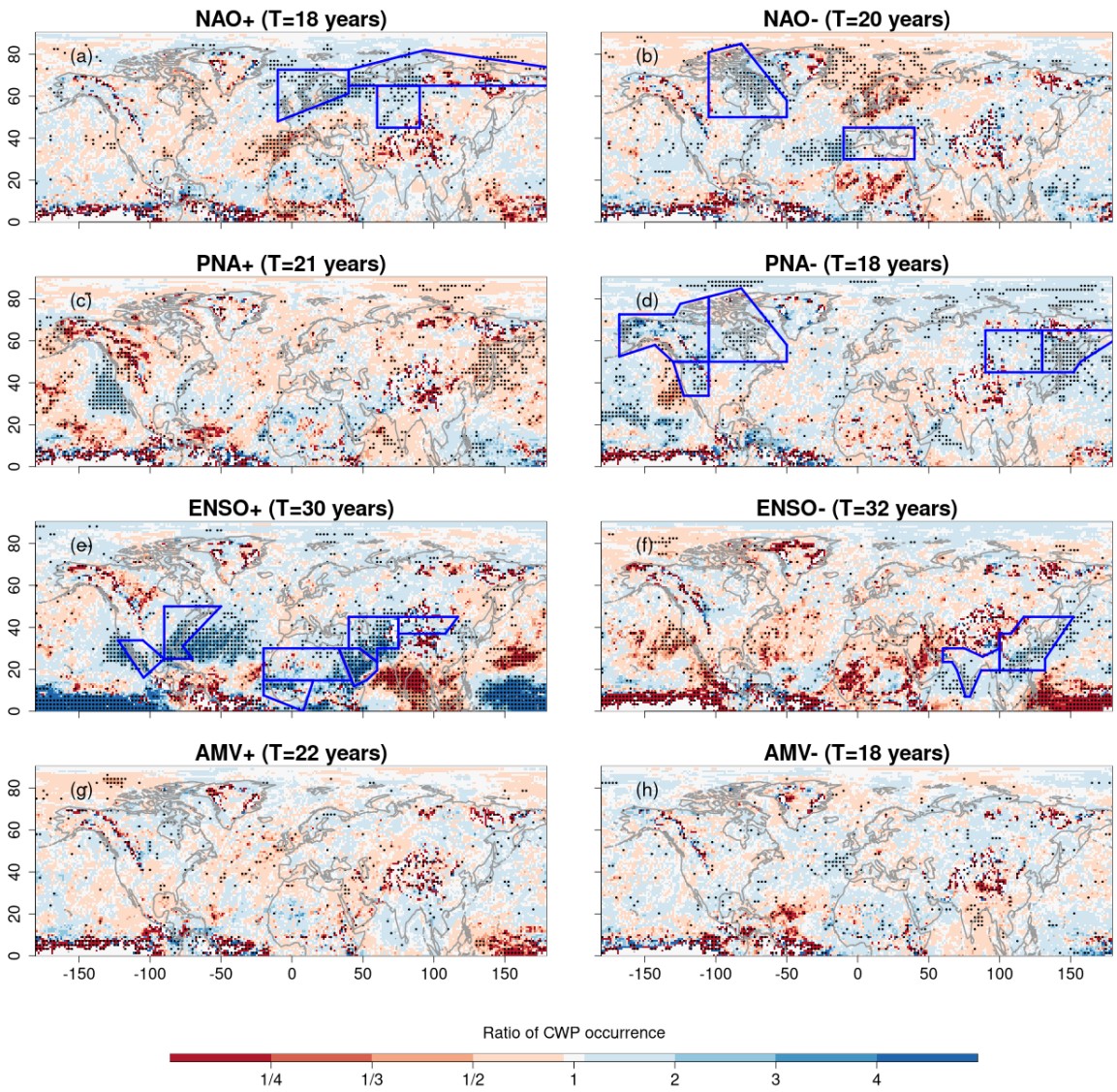

**Figure 2. Direct effect of variability modes on wintertime compound wind and precipitation (CWP) extremes.** Ratio of the average wintertime (December-February) CWP extreme frequencies for the (a) positive and (b) negative phases of NAO (while other variability modes are in their neutral phases) compared to neutral conditions (all variability modes being in their neutral phases) based on the CESM model. Corresponding maps are also displayed for (c, d) PNA, (e, f) ENSO and (g, h) AMV. Numbers in the headers indicate the conditional empirical return period T (in years) for positive and negative phases of the variability modes (while other modes are in their neutral phases). The empirical return period for neutral conditions (i.e., when all modes are in their neutral phases) is T=3 years. Stippling indicates significant differences of mean frequency relative to neutral conditions at the 10% significance level using permutation tests (two-sided) with Bonferroni correction. The framed regions in blue are those where the direct effects of variability modes significantly increase regionally averaged CWP extreme frequencies compared with neutral conditions (following the methodology defined in subsection 2.2.3, and reflecting information in Fig. 3).

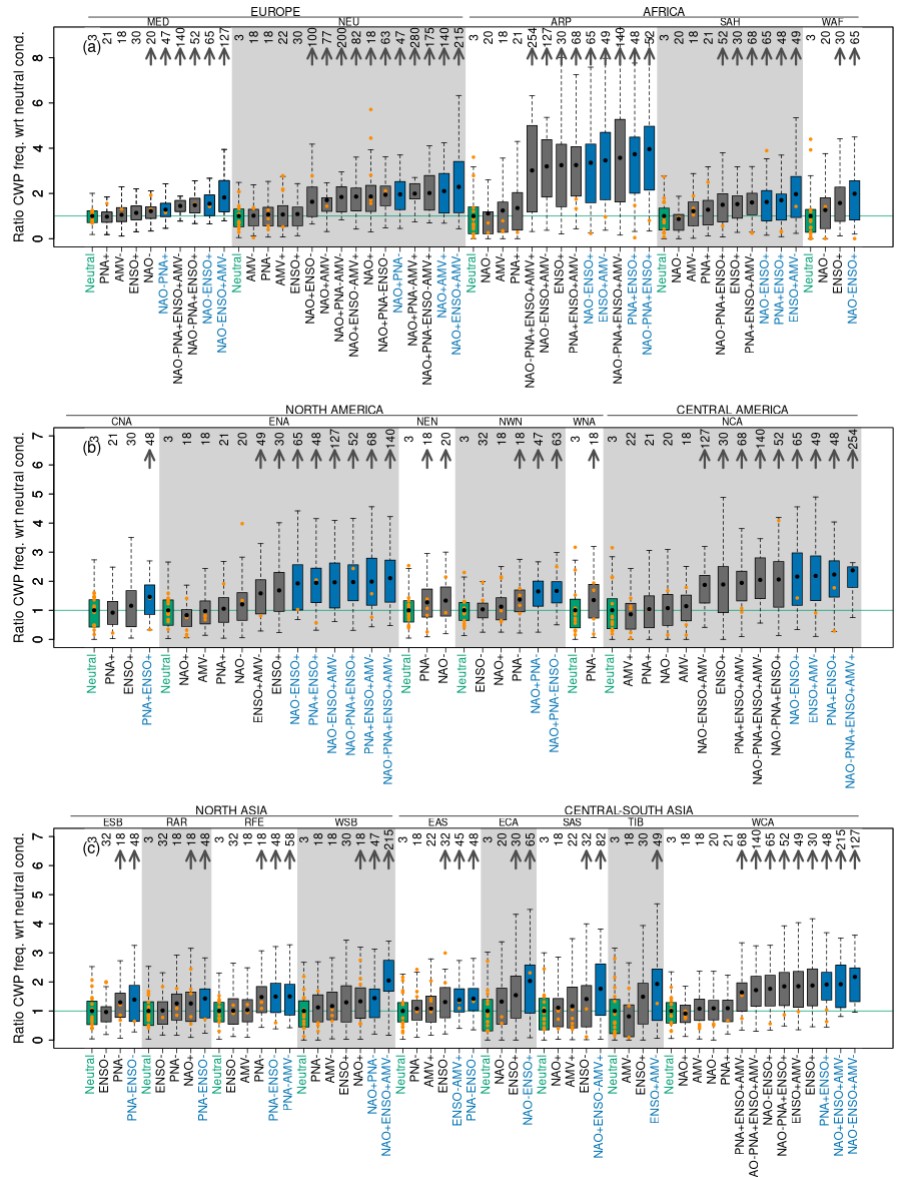

**Figure 3. Influence of concurrent variability modes on regional compound wind and precipitation (CWP) extremes.** Ratio of the regionally averaged wintertime CWP extreme frequencies for the relevant individual and concurrent variability modes (following the methodology defined in subsection 2.2.3) with respect to average frequency for neutral conditions based on the CESM model (orange dots represent associated values of the ratio derived from ERA5, when available, with each dot representing one winter season). Results are displayed for (a) Europe and Africa, (b) North America and Central America, and (c) Northern Asia and Central-South Asia. The conditional empirical return periods (in years; see Methods) for individual and concurrent variability modes are indicated on the top. Grey arrows indicate significant differences in the mean frequency relative to neutral conditions (at the 10% significance level, with Bonferroni correction). Green boxes indicate distributions for neutral conditions (all variability modes being in their neutral phases) and green horizontal lines indicate ratios equal to one. Blue boxes indicate mode combinations with amplified effects, that is higher means of regionally averaged CWP extreme frequencies than their underlying mode sub-combinations (methods subsection 2.2.3). We remove concurrent variability modes that occur in less than 10 winters and regions without significant effects arrows (EEU, WCE, GIC, CAR and SCA).

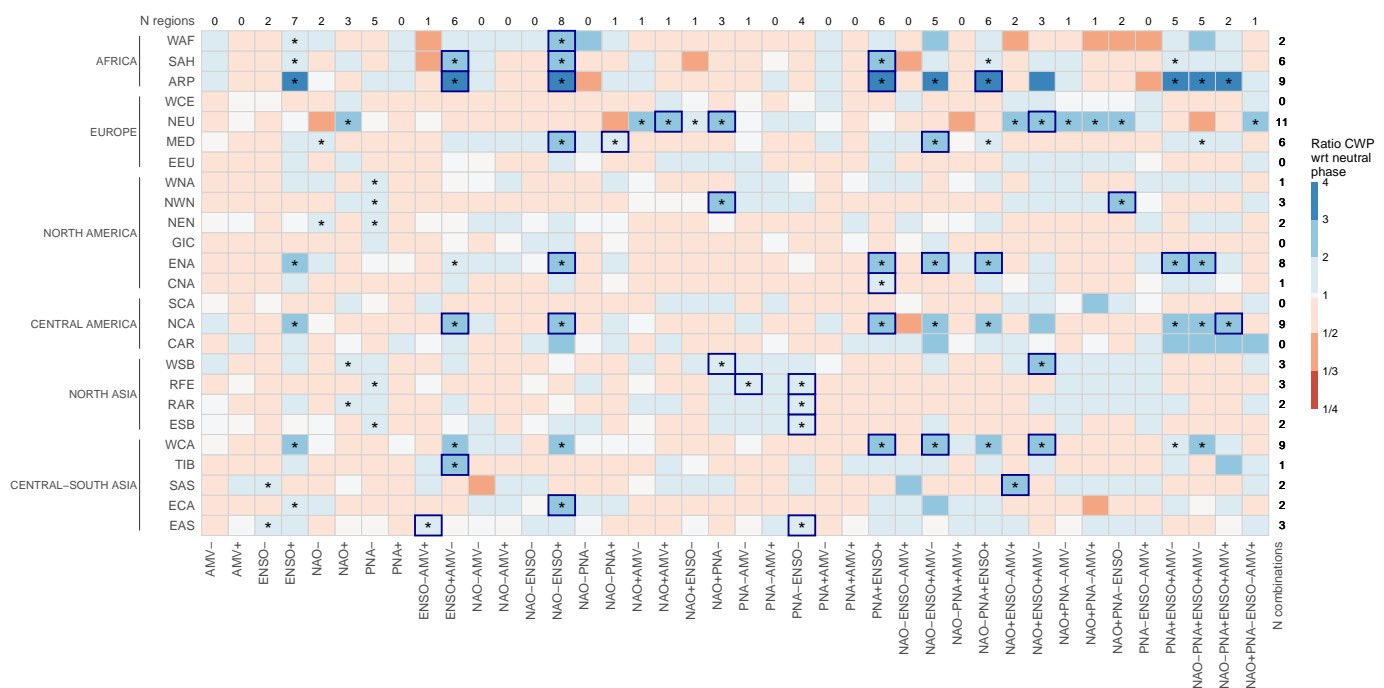

**Figure 4. Effects of variability modes on wintertime compound wind and precipitation (CWP) extremes in SREX regions.** For different regions (see labels on the left side of the matrix), the ratio of the regionally averaged CWP extreme frequencies for individual and concurrent variability modes (see labels on the bottom side of the matrix) with respect to the average frequency under neutral phases of all modes (Metric 1, see Methods), based on the CESM model. Note that positive effects of concurrent variability modes (ratio above 1) tend to be stronger than negative effects (ratio below 1). The individual and concurrent variability modes with significant effects (at the 10% level with Bonferroni correction) relative to neutral conditions are indicated with an asterisk. Blue cell borders indicate combinations with higher means of regionally averaged CWP extreme frequencies than their underlying mode sub-combinations (methods in subsection 2.2.3). To ensure robustness, we do not display concurrent variability modes that occur in less than 10 winters. Based on the displayed information, the numbers on the top and right margins indicate the total count of regions and mode combinations with significant effects of modes, respectively.

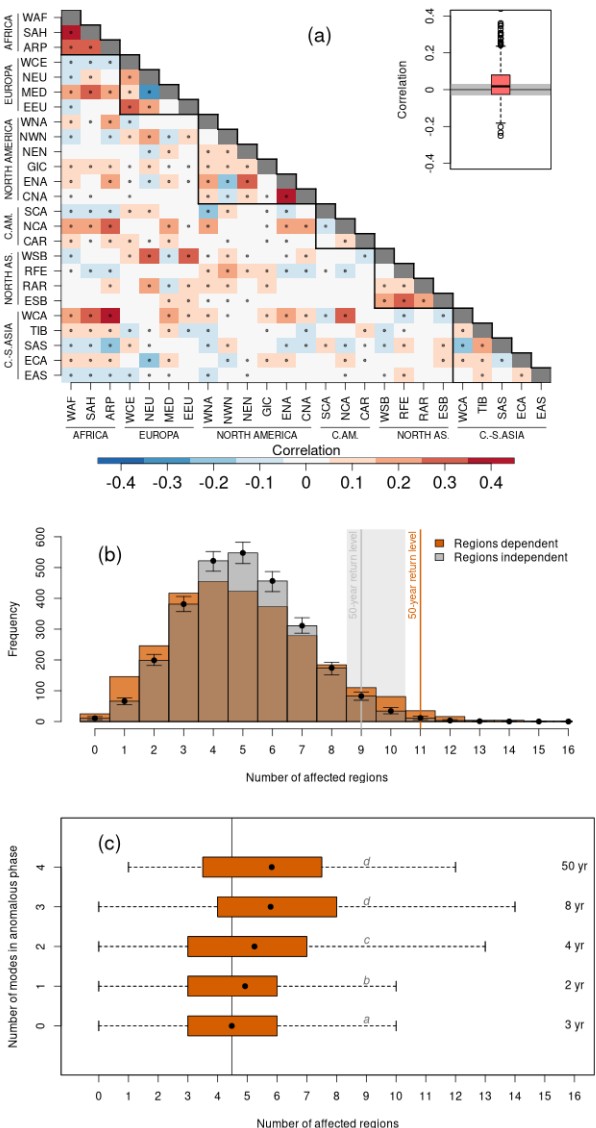

**Figure 5. Dependencies between regional wintertime compound wind and precipitation (CWP) extremes.** (a) Matrix of Spearman correlations for regionally averaged wintertime (December-February) CWP extreme frequencies between regions based on the CESM model. Stippling indicates significant correlations (correlation values that are outside the 90% centred confidence interval). A contour is added for regions in the same macroarea. The boxplot shows summaries of the distribution of the correlations shown in the matrix (interquartile range, median and outliers); the grey background shows the confidence interval for no correlation (bootstrap-based 90% range). (b) Histograms of the number of regions with a high frequency of CWP extremes during the same winter based on the CESM model (orange histogram) and when assuming independence between regional wintertime CWP extreme frequencies (grey histogram; obtained via shuffling the data 1000 times via bootstrap). Vertical lines show the 50-year return levels under dependence and independence. Bootstrap-based confidence intervals for each bin and 50-year return level at 10% significance level are also shown. (c) Boxplots of the number of affected regions given different numbers of variability modes in anomalous phases. The conditional empirical return periods (in years; see Methods) for the different number of variability modes in anomalous phases are indicated on the right.**32** Mean of the ratios under a different number of modes in anomalous phases that are not significantly different ($\alpha = 0.10$, one-sided permutation test) are indicated with the same letters (Jiang et al., 2024). The average number of regions affected under neutral conditions is indicated by a vertical line.

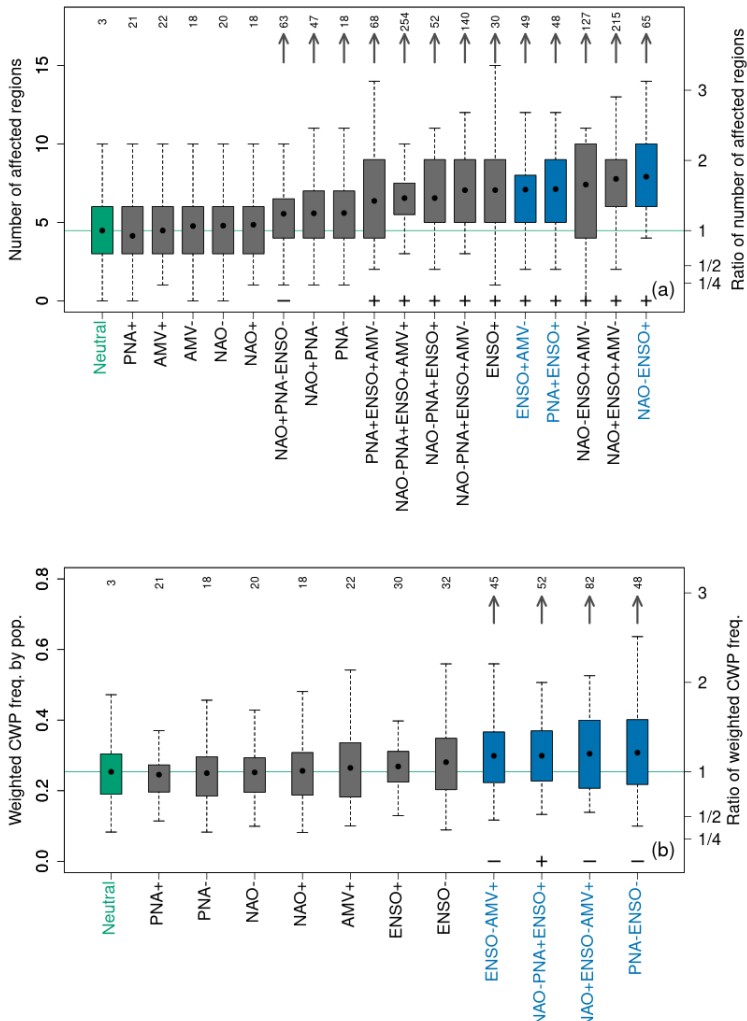

**Figure 6. Influence of variability modes on spatially compounding wind and precipitation (CWP) extremes.** (a) Distributions of the number of regions (left y-axis) experiencing a high frequency of CWP extremes during the same winter (December-February) for different individual and concurrent variability modes (x-axis), based on the CESM model. The right y-axis shows the ratio of the metric to its average under neutral phases of all modes. The combinations presented are selected according to the methodology defined in subsection 2.2.3. Grey arrows indicate significant differences in the mean with respect to neutral conditions at the 10% significance level using one-sided permutation tests and Bonferroni correction. Green boxes indicate distributions for neutral conditions (all variability modes being in their neutral phases). Blue boxes indicate combinations exhibiting higher means than their underlying mode sub-combinations (methods subsection 2.2.3). (b) The same as panel (a), but for the population-weighted CWP extremes over the Northern Hemisphere. Symbols + and − above the x-axis in panels (a) and (b) highlight combinations having a significant effect for a metric containing ENSO+ and ENSO-, respectively.

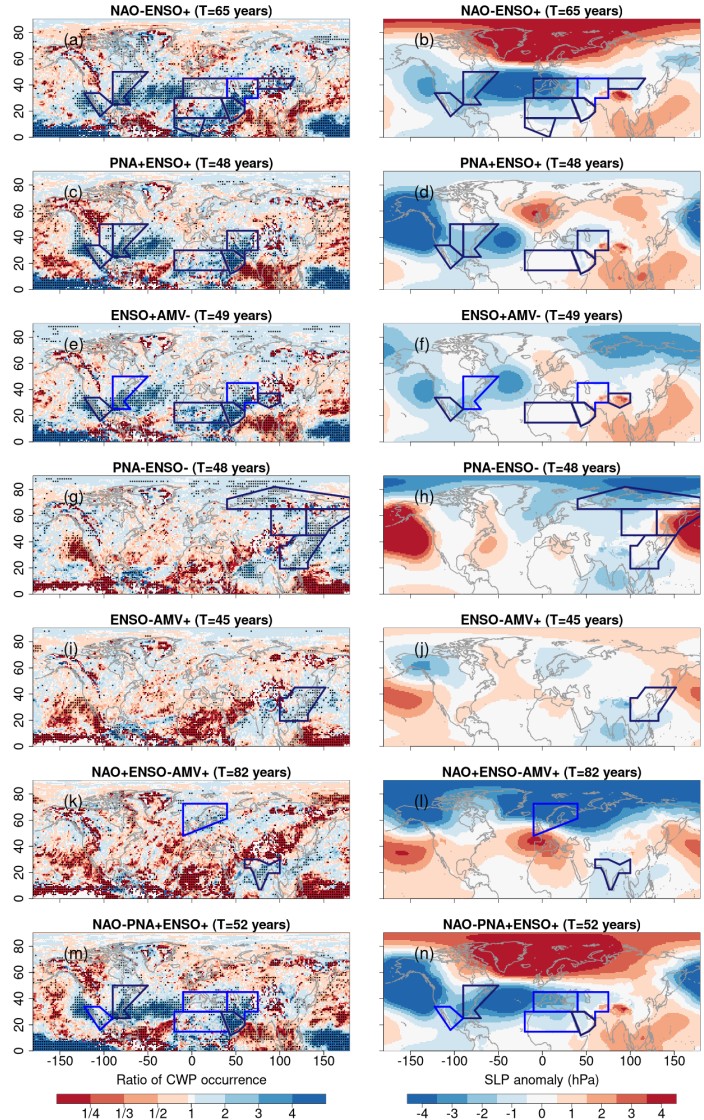

**Figure 7. Compound wind and precipitation (CWP) extremes for concurrent anomalies in variability modes and associated sea level pressure anomalies.** (a) Ratio of the average wintertime (December-February) CWP frequencies for the concurrent variability mode NAO-ENSO+ (while other variability modes are in their neutral phases) compared to winter with all variability modes in neutral phases, based on the CESM model. (b) Mean sea level pressure anomalies for NAO-ENSO+ (while other variability modes are in their neutral phases) compared to neutral conditions (all variability modes being in their neutral phases) based on the CESM model. Corresponding maps are also displayed for (c, d) PNA+ENSO+, (e, f) ENSO+AMV-, (g, h) PNA-ENSO-, (i, j) ENSO-AMV+, (k, l) NAO+ENSO-AMV+ and (m, n) NAO-PNA+ENSO+. Numbers in the headers indicate the conditional empirical return period T (in years; see Methods) for the different concurrent variability modes stated in the title of the panels, whereas the return period for neutral conditions (all modes are in their neutral phases) is T=3 years. Stippling indicates significant differences in mean frequency relative to neutral conditions at the 10% significance level using permutation tests (two-sided) with Bonferroni correction. The framed regions are those where concurrent variability modes significantly increase regionally averaged CWP extreme frequencies compared with neutral conditions; by reflecting information in Fig. 3, the dark framing indicates an amplified effect with respect to underlying mode sub-combinations (methods in subsection 2.2.3).