# Peer review of "Concurrent modes of climate variability linked to spatially compounding wind and precipitation extremes in the Northern Hemisphere"

_EGUsphere, 2024_

## Referee Comment (RC1)

Egusphere-2024-2079

**Concurrent modes of climate variability linked to spatially compounding wind and precipitation extremes in the Northern Hemisphere**

Francois *et al*

**Comments to the authors**

The paper assesses wintertime, northern hemisphere modulation of the frequency of joint precipitation and wind events by four modes of variability. ENSO is found to be the most influential mode, combinations of modes are asserted to essential for the co-occurrence. A doubling of the area affected by joint events is highlighted in the abstract. The analysis is probably sound although it is not currently possible to be certain, and it is currently not clear whether or not it is novel work as the Abstract and Introduction do not convincingly define the research gap being filled. Yet, it is entirely plausible that the gap is real. The manuscript would also benefit from substantive rewriting to improve its clarity.

In short, I feel this has the potential to be an interesting paper if it is very significantly re-written, simplified and/or focussed and clarified.

Major concerns

1. As written, the abstract does not acknowledge any previous work linking joint extremes of precipitation and wind to modes such as ENSO. Linking extremes to ENSO and similar has been a common thing to do for at least 40 years, singularly and jointly. The abstract should be re-written to clearly express/define the research gap with respect to the existing literature. Also note that the use of multiple modes to understand the occurrence of extremes, particularly hurricanes, has a large literature (e.g. Vecchi, 2014). In short, by the end of the Introduction, it is still unclear what the novelty of the paper is (i.e. what is being done that has not been done before). To be clear, there could be a gap (my knowledge of the literature is not comprehensive) but the authors have not convincingly stated what it is. Some papers that might be relevant in the re-framing are:

   • Khouakhi (2017) *Contribution of tropical cyclones to rainfall at global scale*. i.e. ENSO and extreme rainfall.
   • Bloomfield (2024) *Synoptic conditions conducive for compound wind-flood events in Great Britain in present and future climates*. Rain/flood and wind compound and NAO.
   • Hillier (2020) [already cited] *Multi-hazard dependencies can increase and decrease risk*. Flooding and wind compounding and NAO.
   • Vecchi (2014) *On the seasonal forecasting of regional tropical cyclone activity*. Introduction contains literature on use of multiple modes.

2. As certain key pieces of information are not prominent early on, the manuscript (up to Results) can currently only be understood upon re-reading. Please rectify this.
3. Throughout, more care should be taken to ground the work in related literature. Some parts are well referenced, but others are lacking (illustrative examples below).
4. The manuscript lacks a substantive discussion, e.g. about issues relating to the key findings of the paper. Perhaps, this will become clearer when the research gap and focus of the paper is more clearly defined. Perhaps separating the material in the Results into a descriptive (Results) and explanatory (Discussion) parts would help, and clarify the themes the authors wish to discuss.

Detailed Comments

Abstract:

L11 – The phrase '*combinations of modes are essential for the occurrence*' is problematic. They might be necessary for a better description of the co-occurrence from this particular statistical viewpoint, but they are large-scale indicators of conditions, not processes that drive co-occurrence or not. So, please rephrase to a more precise statement.

Introduction:

L25 – 'co-occurring compound' – this is tautology, remove one word.

L26 - (Jeong et al., 2020), this is not the only reference for this. Add 'e.g.' or/and cite another couple.

L44-45 (and throughout the manuscript) – Again, please use 'e.g.' when various papers could be cited. The manuscript should be improved by pairing an older reference with each of the recent singular references used. For example, convective storms were known to produce multiple hazards long before Dowdy & Catto in 2017.

L56 – Why is an AMV abbreviation used here, but is the only mode in the abstract that is not abbreviated. Please be consistent.

Methods:

L91: Daily data used. The focus on this timeframe should be made prominently and clearly (e.g. in the Abstract). [Returning to this, I suggest my comment also highlights that the explanation needs to be clarified]

L98-104: This is unclear, I'm afraid (i.e. not reproducible). E.g. is NAO the mean of DJF, and if so, which days is it applied to – the Jan-Dec year? Please clarify.

L106: First explanation that winter is defined here as DJF. This is a key piece of information, and should be prominent (e.g. in the Abstract). Also, a brief explanation of why DJF is selected is needed.

L106: First indication that this analysis is based on seasonal counts. Again, this should be prominent, because not knowing it leads to a need to re-read the sections above. Please fix this to improve the readability of the manuscript.

L109: Why the 98th percentile?

L111: 'more robust evaluation' – this needs to be more specific please.

L115: It would be beneficial to set the choice of this metric (i.e. a count-based approach e.g. Hillier 2015; Bevaqua 2021 – Guidelines paper; Owen et al 2021 in Weather & Climate Extremes, $\chi$) in the context of the metrics/timescales in previous work (e.g. Hillier & Dixon, 2020; Bloomfield, 2023 in Weather & Climate Extremes)

L119: Two other metrics are 'introduced'. Although I cannot recall exact papers, I struggle to believe this is the first time these sorts of metrics have been used. Again, please place in context of similar metrics and usages with a few references.

L139: 'some confounding effect may remain'. This is a rather important statement. It is good that the authors acknowledge it, however the key question is: How much, and does this impact the key results of the paper? Either by testing with simulated/idealised data, or perhaps another statistical method, I believe that the authors need to answer this question.

L151-152: Scope limiting statement. Fair enough, but I believe this needs prominence in the paper, and am hoping it'll be so in the Discussion at least. It might also need to be in the abstract as it is a potential bias on all conclusions drawn, and so should be prominent for clarity.

Section 2.2.4 – This approach seems reasonable, and statistically significance testing is critical in a paper like this, although I'd need to do a really careful read in a revised manuscript. Illustratively, the permutation procedure would need to account for dependency / relationships between the modes, or statistical significance of any results could be over-estimated (i.e. appear significant when they are not). And, whether this has been done is not currently clear to me.

Results:

L211: biases w.r.t ERA5. Fair enough, although I expect any relevant ones to be explicitly referred back to and results interpreted in light of this during the Discussion.

Section 3.1 & 3.2: From a reader's point of view, it would be nice if this were significantly shorter, drawing out the main points of interest (i.e. that are new).

Please review Section 3.1 as in a number of places it starts to discuss / explain the results to a level that is at or above the limit expected in a Results section.

L257 – 'we move to discussing' Please do not move to discussing in the results section. Please discuss in the Discussion.

L315-326 – This seems like an expansion of or repeat of Methods.  Consider moving to methods.

L336-7 – This long-distance correlation is interesting.  It is an example of the type of thing that could be expanded upon and discussed in a Discussion.

L340 – '*we find that dependencies among regions overall enhance the potential for spatially compounding events*' I am unsure how you can make this conclusion given that you were explicit earlier about only looking at enhancement not reduction of co-occurrence.  Surely, both need to be looked at to comment on an overall effect.

L341 – This, and similar mentions of methodology in Results, should be put into Methods please.

L366 – '*causal links among climate variability modes and oceanic modes exist*'

Discussion:

L425-436 These are assertions, picking highlights from the results.  These results are not discussed, i.e. reflected upon and put in the context of the literature.  Suggest removing, or including in the Results.

L437 – 445: Is a justification of the Methods, which I think is a repeat from the Methods section. Remove.

 L456 – This paragraph is a restatement of the approach, until L456 where an alignment with existing results is stated. So, it would be good to clarify what the new insights provided by this paper are.

L461 – This paragraph is a caveat, which is OK, but should come after a substantive discussion.

Conclusions:

The conclusions are suitable in style, but are difficult to comment while the assertions being made have not previously been discussed.

Fig. 6 – It's good to see the Bonferroni correction being used.

---

## Referee Comment (RC2)

**Review of "Concurrent modes of climate variability linked to spatially compounding wind and precipitation extremes in the Northern Hemisphere"**

**Overall impression of article**

I think it is an important study on compounding extreme events, especially when linking this to potential impact (e.g. through the population metric used in this study). I also value the link to global drivers and teleconnections, as this can benefit short term predictions but also long term projections. Generally, I think this paper needs a bit more restructuring, notably the result and discussion sections. Furthermore, it needs another careful read through because it was difficult at times to understand the sentences. Below I mention more details in some major and some minor suggestions.

**Major points of discussion**

1. The motivation for this study is not so clear to me. Why look at compound wind&precip?
2. Similarly, the motivation for these exact modes of variability is a bit lacking in introduction. You do mention the Indian ocean as a potential influence in discussion. What about other modes? Why not include those?
3. Why do you choose to average the daily wind and precipitation values instead of taking the maximum wind speed and the sum of total precipitation of the day? Especially when it comes to wind, I'm worried that averaging is not the best choice to catch wind-extreme events.
4. Why are you considering seasonal mean indices? Why not look at weekly/monthly data? I think there is an issue with the different timelags here, since ENSO is clearly a yearly oscillation, but the NAO can also be defined on weekly/monthly timescales. I think this has to be motivated from a physical point of view.
5. Which threshold do you end up choosing? It is a bit unclear, you take 95$^{th}$ in ERA5 and 98$^{th}$ in CESM? How do these two compare to each other (I believe you compare 95$^{th}$ in both era and CESM in the supplementary)? Did you do sensitivity experiments to determine these two thresholds are the same?
6. I think the result & discussion sections should be re-structured: you already discuss the findings with respect to other literature in the results, I believe this should be moved to the discussion. In the results only mention your own findings. This will also make your paper easier to read.

**Minor suggestions**

Abstract

1. I miss the motivation for these specific SST-modes of variability in the abstract.
2. In the abstract I had to read the following sentence a few times before I understood: "we identify dependencies enabling extreme spatially compounding events with many regions experiencing CWP extremes in the same winter" L9/10.
3. "mitigation of spatially compounding CWP extremes." L15 how could these CWP extremes be mitigated ?

Introduction

4. L24: "co-occurring compound wind and precipitation (CWP) extremes" co-occuring and compound is that not the same?
5. Introduction: are there any examples of spatially compounding CWP events that lead to extreme damages? You mention the flooding as an example. But it is not entirely

clear to me why CWP should specifically be investigated over other multi-hazard events (hot-dry, no wind-cold, etc.)

6. L46: "cyclones are particularly exposed to CWP extremes" are cyclones not considered a CWP extreme? How is a CWP defined actually?
7. Why do you focus on wintertime CWP only? Aren't summer storms especially damaging (due to trees being in full leaves).
8. L156 how do you calculate significance?

Methods

9. Metric 1: if you average the count per grid point do you still need the latitude weight?
10. Metric 2: why 80$^{th}$ percentile?
11. Why do you only look at positive cases, e.g. when a mode has a positive effect? L151
12. Have you tried any kind of regression analysis? Maybe this also can take away the effect of 'neutral' states not really being neutral, as mentioned L139-140
13. Why not take significance level of 0.05? L 180: *significance level α = 0.10*

Results
14. Your maps would be easier to interpret if you mask out the non-land areas.
15. What's the difference between the following two statements in section 3.2 L250 and L259: *"Model simulations (CESM) show that not only individual variability modes can have effects on regional wintertime frequencies of CWP extremes, but also combinations of modes."* vs *"Model simulations (CESM) show that concurrent anomalies in variability modes amplify the effects of individual modes in many regions."* I think this section needs more attention. There are so many details in the figure, and the text is not complimenting this enough. It is very difficult to understand the main results at the moment, also because you weave discussion in here.
16. L118-119: *"in general agreement with existing literature,"*; either mention the literature or do not mention this. Generally, I think this should be part of the discussion not the results.
17. L315-325 suits better in discussion?
18. L328-331 this is motivation, should maybe go to introduction.
19. L333: *"Figure 5a shows Spearman correlations of regionally averaged CWP extreme frequencies (Metric 1) between all pairs of regions"* You regionally average CWP extreme frequencies, but I'm thinking this could be slightly problematic. The regions are quite large, whereas these CWP extremes can be very local. What happens when you sum the CWP counts instead? Also, why not perform a spatial-dependency analysis on the original CWP data on high frequency, e.g. monthly?
20. L374-376: *"In particular, variability modes in isolation do not lead to significant effects on the population exposure compared to neutral conditions, indicating the importance of considering combinations of modes to distil the effects of modes of variability on the population affected."* Where do you draw this conclusion from? To me it is unclear how this is related to fig 6a (which you reference the sentence before).
21. Fig 7b: but NAO is an index of SLP, so in this sense when you compare NAO- to NAO neutral you will of course find a difference in SLP. Here you go into discussion how a NAO- can physically lead to more CWP extremes *as discussed in other literature.* This should not be a result in my opinion, unless you have actually show a physical mechanism in your results (e.g. convection anomalies, wind anomalies, latent heating anomalies,…). I think this last section of the results is mostly repetition from the

previous sections and can be taken out. Instead focus on interpreting these physical mechanisms in a discussion section.

Discussion
22. It is important to mention you use a climate model in the first sentence already
23. Some sentences are unclear, e.g. L 428: *"Simulations show that extreme spatially compounding events with many regions under CWP extremes in the same winter are enabled by positive dependencies between CWP extremes across different regions"*
24. L339: *"Our model evaluation against ERA5 reanalysis data indicates that the simulated anomalies in CWP extremes associated with modes of variability are well suited for the purpose of our analysis (Figs. S2-S5)"*.
    In my opinion there's some differences between the ERA5 and CESM figures; notably, ERA5 seems more pronounced. There are also regions where ERA5 does not agree with CESM: e.g. S2 shows that parts of North America have a negative ratio in ERA5 under NAO+ whereas this is positive in CESM, or S3 shows parts of North Africa have differences for ENSO+. I think it is important to highlight this, because that means that for some regions we can not make strong statements.
25. Why didn't you include IOD if you mention this has influence on CWP extremes? To me this comes back to the general motivation for this study; the choice for these exact modes need to be motivated clearly.

---

## Referee Comment (RC3)

Concurrent modes of climate variability linked to spatially compounding wind and precipitation extremes in the Northern Hemisphere *(François et al.)*

**Overview:**

This paper focuses on compound wind and precipitation (CWP) extremes, aiming to identify the drivers behind the occurrence of these events in the Northern Hemisphere. Climate model simulations from the Community Earth System Model are used with reanalysis data (ERA5) providing a "sense check". A few key climate variable modes are considered (ENSO, AMV, NAO & PNA). The individual effects of these events are found to follow existing literature, e.g. NAO+ increasing CWP extremes in Northern Europe. Concurrent phases of variability modes are considered with specific regional effects discussed. The NAO- & ENSO+ combination increased the likelihood of CWP extremes in eight regions. This motivated exploring spatially compounding extremes, where a positive trend between the number of anomalous variability modes and the number of regions was identified. Physical mechanisms for the statistical relationships were then discussed. This paper concludes ENSO is the most influential mode of variability for CWP extremes in the Northern Hemisphere.

Compound events are an area of current interest and this manuscript will appeal to the community. It is suitable for this NHESS special issue and I therefore recommend it's publication subject to the changes outlined below. I would therefore appreciate the author's response on the comments below.

**General comments:**

As this study covers a large region and many combinations of variability modes, the presentation of results is important. The paper has a wide scope which at times means detail on specific regions is lacking. Choosing two or three regions or one teleconnection index to focus on gives this study more impact.

While the standard of written English is fine, the language used makes this paper difficult to read at times. There are some very long sentences which could be split up or multiple sentences which may be more readable as a bullet pointed list. Redrafting Section 3 will make the paper more readable and therefore accessible to the wider scientific community.

Figures are meant to help convey information simply, Figures 3, 4 & 5 are complex. The authors should only include combinations of variability modes discussed in the text with the full figures available in the supplementary material.

The choices of percentile thresholds are arbitrary. The results of this study would hold more weight if a sensitivity analysis on these had been conducted. e.g. 98th percentile of daily precipitation seems low as this data is zero inflated.

Daily precipitation is not always proportional to any resulting impact – the authors should acknowledge the complexity of the precipitation-flood relationship. For more on this see Bloomfield et al. (2023) [ https://doi.org/10.1016/j.wace.2023.100550 ].

While compound wind-precipitation events cause large impacts, they are rare (e.g. Fig. 2 from Jones et al. (2024) [ https://doi.org/10.1002/wea.4573 ] ). Considering these extremes in isolation gives the complete picture of a compound hazard. You have cited Manning et al. (2024) to highlight extratropical cyclones as drivers of CWP events, but Manning et al. (2024) notes CWP events can be driven by precipitation extremes.

Concurrent modes of climate variability linked to spatially compounding wind and precipitation extremes in the Northern Hemisphere *(François et al.)*

**Specific comments:**

L2: Change "agricultural crops" to "crops"

L6: Remove NAO & PNA abbreviations, they are not used in rest of abstract.

L13: Remove "For example" here, the reader knows you're giving them an example.

L17-22: Split into two sentences and rejig. Define compound events first, then highlight their importance from this IPCC report.

L51: Useful to describe what the deviation from mean NAO conditions is, how does it affect frequency & intensity of events?

L74: Specify which months the winter season covers.

L75: Change "effective" to "influential"

L84: Make the rationale behind the choice of these regions clearer. These shapes cut across country boundaries, making this study less applicable to the insurance industry.

L96: Why did you chose to begin with 1959? ERA5 covers from 1940 so matching the same period as CESM makes sense.

L96: "Singh et al. (2021)" reference doesn't make sense here? As far as I can tell, Singh et al. (2021) doesn't use ERA5?

L110: The 95$^{th}$ percentile of daily data considers 1114 days in this period (1959-2019) to be extreme. Yet the 98$^{th}$ percentile over 1950-2019 only considers 511 extreme days. Surely a higher threshold of ERA5 data is required for these periods to be comparable?

L115: Include rationale for weighting by cosine of latitude.

L151: Change "That is, in this study, we do not..." to "This study does not".

L154: Remove ", in principle,"

L162: The 280 year return period seems to be an arbitrary choice. Sensitivity analysis on this threshold would be of interest.

L176: Mismatched bracket after "subsection 2.2.3".

L180: A 10% significance level seems high, 5% (or even 1%) level is much more standard practice.

L185: How many times is "several times"? State this in the text.

L199-200: Change 100.000 to 100,000

L224: A significant body of literature exists linking extreme windstorms to strong winds (favourable conditions for CWP events). Here I would at least cite:

- Mailier et al. (2006) https://doi.org/10.1175/MWR3160.1
- Priestley et al. (2024): https://doi.org/10.5194/nhess-23-3845-2023

L312: I'd make this sentence clearer, "generally covers most of the time" is very ambiguous.

Concurrent modes of climate variability linked to spatially compounding wind and precipitation extremes in the Northern Hemisphere *(François et al.)*

L391: Change "Europa" to "Europe".

L430: Is this not driven by atmospheric circulation patterns?

L480: Change "found" to "estimated"

L484: A natural next step would be repeating this study for the southern hemisphere.

---

## Author Comment (AC1)

**Response to Referee Comment 1: "Concurrent modes of climate variability linked to spatially compounding wind and precipitation extremes in the Northern Hemisphere"**

**Comments:**

The paper assesses wintertime, northern hemisphere modulation of the frequency of joint precipitation and wind events by four modes of variability. ENSO is found to be the most influential mode, combinations of modes are asserted to be essential for the co-occurrence. A doubling of the area affected by joint events is highlighted in the abstract. The analysis is probably sound although it is not currently possible to be certain, and it is currently not clear whether or not it is novel work as the Abstract and Introduction do not convincingly define the research gap being filled. Yet, it is entirely plausible that the gap is real. The manuscript would also benefit from substantive rewriting to improve its clarity. In short, I feel this has the potential to be an interesting paper if it is very significantly re-written, simplified and/or focussed and clarified.

**Response:**

We would like to thank the referee for their positive comments and the detailed questions. All the comments and our point-by-point responses are given below.

**Comments:**
**Major concerns**
1. As written, the abstract does not acknowledge any previous work linking joint extremes of precipitation and wind to modes such as ENSO. Linking extremes to ENSO and similar has been a common thing to do for at least 40 years, singularly and jointly. The abstract should be re-written to clearly express/define the research gap with respect to the existing literature.

**Response:**
We thank the referee for this comment. We modified the Abstract and Introduction. Please find below the modifications (in blue) of the Abstract (starting at L1 of the initial submitted article) related to this comment.

**L1:** "Compound wind and precipitation (CWP) extremes often cause severe impacts on human society and ecosystems, such as damage to crops and infrastructure.  **Spatially compounding events with multiple regions affected by CWP extremes in the same winter can impact the global economy and reinsurance industry, however our understanding of these events is limited. While climate variability modes such as El Niño Southern Oscillation (ENSO) can influence the frequency of precipitation and wind extremes, their individual and combined effects on spatial co-occurrences of CWP extremes across the Northern Hemisphere have not been systematically examined.**"

**Comments:**
Also note that the use of multiple modes to understand the occurrence of extremes, particularly hurricanes, has a large literature (e.g. Vecchi, 2014). In short, by the end of the Introduction, it is still unclear what the novelty of the paper is (i.e. what is being done that

has not been done before). To be clear, there could be a gap (my knowledge of the literature is not comprehensive) but the authors have not convincingly stated what it is. Some papers that might be relevant in the re-framing are:
• Khouakhi (2017) Contribution of tropical cyclones to rainfall at global scale. i.e. ENSO and extreme rainfall.
• Bloomfield (2024) Synoptic conditions conducive for compound wind-flood events in Great Britain in present and future climates. Rain/flood and wind compound and NAO.
• Hillier (2020) [already cited] Multi-hazard dependencies can increase and decrease risk. Flooding and wind compounding and NAO.
• Vecchi (2014) On the seasonal forecasting of regional tropical cyclone activity. Introduction contains literature on use of multiple modes.

**Response:**

We thank the referee for this comment and the papers provided for the re-framing of the Introduction. We added the following texts in the Introduction to explicitly clarify the novelty of the paper:

L42 of the article initially submitted: "**Understanding the drivers of CWP extremes is crucial for optimising resource distribution during response efforts, yet research in this area remains limited. Several studies have explored the influence of single or combined mode of variability anomalies on precipitation and wind extremes in isolation, both at regional (e.g., Elsner et al., 2001, Abeysirigunawardena et al., 2009, Kossin et al., 2010, Grimm, 2011) and global scales (e.g., Khouakhi et al., 2017, Gao et al., 2022, Liu et al., 2024). Regarding CWP extremes, some studies have explored the influence of individual variability modes at the regional scale only (e.g., Hillier et al., 2020, Bloomfield et al., 2024), thus not allowing the assessment of the relation between concurrent climate variability modes and spatial co-occurrences of CWP extremes across several regions.**"

L76 of the article initially submitted: "**To the best of the authors' knowledge, the present study is the first to investigate CWP extremes and associated spatially compounding events across the Northern Hemisphere as mediated by multiple large-scale modes of variability. We address this research gap by using** large ensemble climate model…**"

**Comments:**

2. As certain key pieces of information are not prominent early on, the manuscript (up to Results) can currently only be understood upon re-reading. Please rectify this.

**Response:**

We identify 1) the use of *daily* CWP extremes, 2) the focus on the winter defined as *December-January-February months* as partially missing information up to the Results section, 3) the derivation of the metrics at the seasonal time frame using seasonal counts, and 4) the focus on **increases** of CWP extremes and associated spatial co-occurrences. Changes have been made to the Abstract and Introduction to ensure that this information is highlighted correctly and at the right time to avoid re-reading. Additionally, we added a few words to the text (up to the Results section) to clarify this information throughout the reading.

**Comments:**

3. Throughout, more care should be taken to ground the work in related literature. Some parts are well referenced, but others are lacking (illustrative examples below).

**Response:**

Thanks for this comment. We included more literature throughout the article, taking care of the illustrative examples described by the referee below.

**Comments:**

4. The manuscript lacks a substantive discussion, e.g. about issues relating to the key findings of the paper. Perhaps, this will become clearer when the research gap and focus of the paper is more clearly defined. Perhaps separating the material in the Results into a descriptive (Results) and explanatory (Discussion) parts would help, and clarify the themes the authors wish to discuss.

**Response:**

Thanks for this comment. We have rewritten the Discussion Section, in addition to removing elements of discussion that were present in the Results section before, and thus were misplaced.

**Comments:**
**Detailed Comments**
Abstract:
L11 – The phrase 'combinations of modes are essential for the occurrence' is problematic. They might be necessary for a better description of the co-occurrence from this particular statistical viewpoint, but they are large-scale indicators of conditions, not processes that drive co-occurrence or not. So, please rephrase to a more precise statement.

**Response:**

We modified (in blue) the following sentence starting at L10 of the initial submitted article (Abstract):

*"While ENSO is the most influential variability mode for such extreme spatially compounding events,  the occurrence of these events **increases further when multiple modes of variability are in anomalous phases. In particular**, combinations of modes increase both the number of regions and the population exposed to daily CWP extremes in the same winter. **For example, combined ENSO- and NAO+ nearly doubles** the number of affected regions compared to neutral conditions on average."*

**Comments:**
Introduction:
L25 – 'co-occurring compound' – this is tautology, remove one word.

**Response:**

Thanks for this comment. We suggest removing the word "co-occurring".

**Comments:**
L26 - (Jeong et al., 2020), this is not the only reference for this. Add 'e.g.' or/and cite another couple.

**Response:**
Thanks for this comment. We added three additional references:

*"For example, the combination of high winds and rainwater can result in severe damage due to the inflow of water through joints or cracks in building structures (**e.g., Blocken and Carmeliet, 2004; Mirrahimi et al., 2015; Martius et al., 2016,** Jeong et al., 2020)".*

Additional references added:
- Blocken B, Carmeliet J (2004) A review of wind-driven rain research in building science. J Wind Eng Ind Aerodyn 92(13):1079–1130
- Mirrahimi S, Lim CH, Surat M (2015) Review of method to estimation of wind-driven rain on building facade. Adv Environ Biol 9(2):18–23
- Martius, O., Pfahl, S., and Chevalier, C.: A global quantification of compound precipitation and wind extremes, Geophys. Res. Lett., 43, 7709–7717, https://doi.org/10.1002/2016GL070017, 2016.

**Comments:**
L44-45 (and throughout the manuscript) – Again, please use 'e.g.' when various papers could be cited. The manuscript should be improved by pairing an older reference with each of the recent singular references used. For example, convective storms were known to produce multiple hazards long before Dowdy & Catto in 2017.

**Response:**
Thanks for this comment. We added '**e.g.,**' to cite papers when appropriate. Also, we paired older references with recent singular ones, when possible. The list of new citations to answer this comment is provided below:
- L43: Ralph et al. (2006) for the links between atmospheric rivers and extremes.
- L45: Rappaport et al (2000) for the links between low-pressure systems and extremes.
- L45: Cerveny et al. (2000) for the links between tropical cyclones and extremes.
- L45: Raible (2007) for the links between extratropical cyclones and extremes.
- L102: Kawamura et al. (2004) for lagged effects of ENSO.
- L177: Bradley et al. (1968) for permutation tests.
- L451: Chongyin et al. (2001) for links between IOD and Asian monsoon.
- L454: Ashok et al. (2003) for the links between IOD and ENSO.
- L473: Kirov et al. (2002) for the interactions between ENSO and NAO.

**Comments:**
L56 – Why is an AMV abbreviation used here, but is the only mode in the abstract that is not abbreviated. Please be consistent.

**Response:**
Thanks for this comment. We added the AMV abbreviation in the abstract, ensuring consistency across modes in the manuscript.

**Comments:**

Methods:
L91: Daily data used. The focus on this timeframe should be made prominently and clearly (e.g. in the Abstract). [Returning to this, I suggest my comment also highlights that the explanation needs to be clarified]

**Response:**
Thanks for this comment. We simply replaced some of the terms "CWP extremes" by "daily CWP extremes" in the Abstract, Introduction and Methods sections to precise the focus of the timeframe.

**Comments:**
L98-104: This is unclear, I'm afraid (i.e. not reproducible). E.g. is NAO the mean of DJF, and if so, which days is it applied to the Jan-Dec year? Please clarify.

**Response:**
Thanks for this comment. We added some clarifications to the text:

**L98 of the article initially submitted:** "Seasonal indices for the two oceanic (ENSO and AMV) and two atmospheric (NAO and PNA) variability modes for both CESM and ERA5 data are calculated from monthly data using the National Center for Atmospheric Research (NCAR) data package Climate Variability Diagnostics Package (CVDP, Phillips et al., 2014). The **seasonal** indices for the NAO, PNA and AMV  **are calculated as the mean of the monthly values for December, January and February**. **For these indices, December is taken from the year *n−1*, while January and February are taken from year *n*, and the resulting seasonal index is assigned to year *n*. For** ENSO we **proceeded similarly, but used November–January averages to account** for lagged effects (Li et al., 2011; Hong Lee et al., 2023)."

**Comments:**
L106: First explanation that winter is defined here as DJF. This is a key piece of information, and should be prominent (e.g. in the Abstract). Also, a brief explanation of why DJF is selected is needed.

L106: First indication that this analysis is based on seasonal counts. Again, this should be prominent, because not knowing it leads to a need to re-read the sections above. Please fix this to improve the readability of the manuscript.

**Response:**
Thanks for this comment. We now mention that winter is defined as December-January-February in the Abstract and Introduction. We also add a brief explanation for this choice, and precise that the analysis is based on seasonal counts.

L42 of the article initially submitted: *"To advance our understanding of CWP extremes and their drivers, this study focuses on the Northern Hemisphere due to the high population density and the severe impacts of CWP extremes in this part of the world (e.g., Liberato et al., 2014, Wahl et al., 2015, Raveh et al., 2015). Such compound extremes are most frequent in coastal regions of the Pacific and Atlantic oceans (e.g.,*

*Maraun et al. 2016) and in the winter season (e.g., Greeves et al., 2007, Hansen et al., 2019). Consequently, this study focuses on seasonal frequency of daily CWP extremes during December-February."*

**Comments:**
L109: Why the 98th percentile?

**Response:**
Thanks for this comment. Using percentiles to define extremes is frequently done in the literature as, for example, it allows to control the sample size for robust statistical analyses (e.g., Zhang et al., 2011, Martius et al., 2016). Although using different variables (that is, wind gusts instead of wind speed), some studies considered the local 98th percentile to investigate precipitation and wind extremes (Martius et al., 2016). Also, Klawa and Ulbrich (2003) show that the local 98th wind percentile is a damage-relevant wind threshold for wind gusts. In our study, daily data for the December-January-February months are used. It represents 90 days by season. By choosing the 98th percentile as a threshold, the expected number of exceedances per season for wind and precipitation in isolation is equal to 90*0.02 ≈ 2 events per season, which can be considered sufficient to analyse co-occurrences of wind and precipitation values above these thresholds. Choosing a percentile higher than the 98th would allow us to focus on more extreme events (e.g., Zhang et al., 2011), but it would reduce the sample size for the analyses. Note that for model evaluation, we use the 95th percentiles, which is needed given the shorter length of reanalysis data. Although CWP events exceeding the 98th percentile of wind and precipitation can be considered moderate extremes, we think that they can still be considered impact-relevant. Choosing a local and impact-relevant threshold for wind and precipitation extremes is difficult, especially for precipitation for which incorporating effects such as surface runoff, snow melt and landslides would be needed (e.g., Williams, 1978) and is thus out of the scope of this study.

We suggest to add the following explanations (in blue) to the text:

**L110 of the article initially submitted:** "We use the 98th percentile of wind and precipitation over the 1950–2019 period for the main analysis based on data from the CESM model. **Percentile-based thresholds are frequently used to investigate climate extremes (e.g., Zhang et al., 2011, Martius et al., 2016). Following Klawa and Ulbrich (2003) and Martius et al (2016), we chose the 98th percentile, which is a compromise to capture the most extreme events in the CESM simulations while ensuring a sufficiently large sample size for robust statistical analysis.** For model evaluation**, which involves both the CESM model and ERA5 reanalyses (Figs. S1-S5 of the Supplement only)**, we use the 95th percentiles over the 1959-2019 period -- such a lower threshold allows for a more robust evaluation. **The reason for this is that,** given the ERA5's limited period, **extremes in the reanalysis data set are more scarce and associated statistics for very extreme events are largely affected by sampling uncertainty (Bevacqua et al., 2021b). Selecting a slightly lower threshold allows us to reduce this sampling uncertainty and thus improve confidence in assessing the model's ability to simulate extremes (e.g., Bevacqua et al., 2021b, Kelder et al., 2022, Fischer et al., 2023)**."

Following the suggestion of the third referee, we produced a sensitivity analysis by considering the 99th and 99.5th percentile to define seasonal counts of CWP extremes. New Figures S16-S19 have been added to the Supplement and display the results we obtained for:
- Metric 1: the influence of individual and concurrent variability modes on regional wintertime CWP frequency (Figs. S16 and S17, same results as those presented in Fig. 3 but consider the 99th and the 99.5th percentile, respectively).

- Metrics 2 and 3: the influence of variability modes on spatially CWP extremes (Figs. S18 and S19, same results as those presented in Fig. 6 but consider the 99th and the 99.5th percentile, respectively).

We added some sentences in the Discussion to summarize the results of the sensitivity analyses. For more details, see our response to comments from the third referee.
"**While the 98th percentile has been used in this study to focus on extremes and is relatively well-established in the literature (e.g., Klawa et al., 2003, Martius et al., 2016), other higher thresholds could have been chosen to consider more intense extreme events (e.g., Liu et al., 2013, Schar et al., 2016, Camuffo et al., 2020). Figs. S16-S19 show results from a sensitivity analysis on the influence of variability modes on regional CWP extremes (Metric 1) and spatially compounding events (Metrics 2 and 3) with the 99th and 99.5th percentiles used as thresholds. Although there are some variations in the results compared to those for the 98th percentile, the main conclusions drawn across the different thresholds are broadly consistent for all Metrics. The magnitude of the effects of the combinations are generally consistent across thresholds, and the combinations detected at higher thresholds are generally included among those identified at lower thresholds (Figs. S16-S19). Such slight differences may be due to larger sampling uncertainty for higher thresholds limiting the ability to detect significant effects for higher thresholds rather than different physical mechanisms involved for different thresholds. While the sensitivity analyses broadly indicate the robustness of most of our findings, possible relevant differences across thresholds highlight the importance of identifying impact-relevant thresholds, though this task is challenging (Williams, 1978, Bloomfield et al., 2023).**"

**References:**
- Zhang, X. B., L. Alexander, G. C. Hegerl, P. Jones, A. K. Tank, T. C. Peterson, B. Trewin, and F. W. Zwiers (2011), Indices for monitoring changes in extremes based on daily temperature and precipitation data, *Wires Clim. Change*, 2, 851–870.
- Klawa, M., and U. Ulbrich (2003), A model for the estimation of storm losses and the identification of severe winter storms in Germany, *Nat. Hazard Earth Syst. Sci*, 3, 725–732.
- Williams, G. P. (1978), Bank-full discharge of rivers, *Water Resour. Res.*, 14(6), 1141–1154, doi:10.1029/WR014i006p01141.

**Comments:**
L111: 'more robust evaluation' – this needs to be more specific please.

**Response:**

We agree with this comment and we modified (in blue) the following sentences starting at L111 of the initial submitted article (Method).

**L110 of the article initially submitted:** "We use the 98th percentile of wind and precipitation over the 1950–2019 period for the main analysis based on data from the CESM model. **Percentile-based thresholds are frequently used to investigate climate extremes (e.g., Zhang et al., 2011, Martius et al., 2016). Following Klawa and Ulbrich (2003) and Martius et al (2016), we chose the 98th percentile, which is a compromise to capture the most extreme events in the CESM simulations while ensuring a sufficiently large sample size for robust statistical analysis.** For model evaluation**, which involves both the CESM model and ERA5 reanalyses (Figs. S1-S5 of the Supplement only)**, we use the 95th percentiles over the 1959-2019 period -- such a lower threshold allows for a more robust evaluation. **The reason for this is that,** given the ERA5's limited period, **extremes in the reanalysis data set are more scarce and associated statistics for very extreme events are largely affected by sampling uncertainty (Bevacqua et al., 2021b). Selecting a slightly lower threshold allows us to reduce this sampling uncertainty and thus improve confidence in assessing the model's ability to simulate extremes (e.g., Bevacqua et al., 2021b, Kelder et al., 2022, Fischer et al., 2023)**."

**Comments:**
L115: It would be beneficial to set the choice of this metric (i.e. a count-based approach e.g. Hillier 2015; Bevaqua 2021 – Guidelines paper; Owen et al 2021 in Weather & Climate Extremes, $\chi$) in the context of the metrics/timescales in previous work (e.g. Hillier & Dixon, 2020; Bloomfield, 2023 in Weather & Climate Extremes)

**Response:**
Thanks for this suggestion. We added (in blue) the following sentences before the explanations on Metric 1 starting at L106 of the article initially submitted (Methods):

L106 of the article initially submitted:
**"2.2.1. CWP extremes**

**Many techniques have been utilized to characterize CWP extremes, with the selection of a specific method being guided by the research question. For example, the correlation between wind and precipitation has been quantified at daily to seasonal timescales (e.g., Matthews et al., 2014, Deluca et al., 2017, Hillier et al., 2020, Bloomfield et al., 2023). Logistic regression models have been applied to quantify the likelihood of a precipitation extreme occurring given the presence of a wind extreme (e.g., Martius et al., 2016). Alternative approaches include examining tail dependence (e.g., Vignotto et al., 2021) and employing impact-focused metrics (e.g., Hillier et al., 2015, Hillier et al., 2020, Bevacqua et al., 2021b). The most straightforward approaches include counting extreme wind and precipitation co-occurrences above a given percentile (e.g., Martius et al., 2016, Bevacqua et al., 2021b), or using extremal dependency measures estimating the probability of one variable being extreme given that the other one is extreme (e.g., Coles et al., 1999, Hillier et al., 2015, Owen et al., 2021). Here, t**o investigate winter season (December–February) CWP extremes at the grid cell level**, we derived seasonal counts of CWP extremes, defined as wind and precipitation

values simultaneously exceeding high thresholds. This results in one count per season per grid cell, which allows for investigating the effect of seasonally-averaged climate variability modes on the counts."

**Comments:**
L119: Two other metrics are 'introduced'. Although I cannot recall exact papers, I struggle to believe this is the first time these sorts of metrics have been used. Again, please place in context of similar metrics and usages with a few references.

**Response:**
These metrics (or similar ones) may have already been used in other studies. Therefore, we replaced the word "introduce" by "use" to avoid readers thinking that these measures are totally new.

For Metric 2, we modified the following sentence (in blue) L125 of the article initially submitted:

L125: "Then, **similar to Singh et al. (2021),** the total number of affected regions during the same winter is counted."

For Metric 3, we added the following sentence (in blue) L131 of the article initially submitted:
L131: "**Population weighting is utilized here as a surrogate for the assets at risk that could experience damages due to CWP extremes (e.g., Bloomfield et al., 2023).**"

**Comments:**
L139: 'some confounding effect may remain'. This is a rather important statement. It is good that the authors acknowledge it, however the key question is: How much, and does this impact the key results of the paper? Either by testing with simulated/idealised data, or perhaps another statistical method, I believe that the authors need to answer this question.

**Response:**
Thanks for this comment. We would like to note that by fixing the non-considered modes to neutral conditions, we devised an approach that allows for taking into account confounding effects arising from the considered modes. We modified the text around this aspect in the new text as we realised that this was not clear enough.

L139: "Following Singh et al. (2021), additionally conditioning all other modes in the neutral phase in (i) **serves** for better isolating the causal effects of the individual variability mode of interest. **Specifically, such additional conditioning allows for taking into account confounding effects arising from the considered modes, still**  some confounding effects may remain. **In particular,** modes in neutral states still vary within the range of neutral conditions and we do not control for them. **In addition,** further effects may arise from variability mode not considered in this study. Similarly to the analysis of single variability modes, we quantify the effect of concurrent variability modes in non-neutral phases based on the ratio between…"

Still, as with any statistical method, some limitations exist, and we wanted to acknowledge this transparently. As in all approaches, confounding effects can also arise from

non-considered variables. While adding other modes could have helped in this direction, a choice on the number of considered modes has to be made, and we considered modes known for having effects on the CWP extremes. Note also that adding extra conditioning variables/modes would reduce the sample size when deriving conditional statistics, therefore increasing uncertainties and decreasing the robustness of the results. We also note that other statistical methods, such as regression, are often used to investigate causal links (e.g., Pearl, 2013), but occurrence-based methods, as we use in our study, can also be considered for such purposes (e.g., Kretschmer et al., 2021).

Overall, while we recognize inherent limitations in studies like ours with respect to r confounding effects, we believe that our study provides an important overview into the effects of the combined drivers of CWP extremes and associated spatial compounding events in the Northern Hemisphere, paving the way for further research into causal effects.

We modified a sentence in the Discussion section:
"**Applying statistical methods such as regression techniques (e.g., Pearl, 2013, Kretschmer et al., 2021), or more advanced approaches such as causal networks (e.g., Nowack2020) may help to shed light on the complete causal pathway leading to spatially compounding events and better control potential confounding effects.**"

References:

- Kretschmer, M., S. V. Adams, A. Arribas, R. Prudden, N. Robinson, E. Saggioro, and T. G. Shepherd, 2021: Quantifying Causal Pathways of Teleconnections. *Bull. Amer. Meteor. Soc.*, **102**, E2247–E2263, https://doi.org/10.1175/BAMS-D-20-0117.1.
- Pearl, J., 2013: Linear models: A useful "microscope" for causal analysis. *J. Causal Inference*, 1, 155– 170, https://doi.org/10.1515/jci-2013-0003.
- Pearl, J., 2000: *Causality: Models, Reasoning and Inference.* 2nd ed. Cambridge University Press, 478 pp.

**Comments:**
L151-152: Scope limiting statement. Fair enough, but I believe this needs prominence in the paper, and am hoping it'll be so in the Discussion at least. It might also need to be in the abstract as it is a potential bias on all conclusions drawn, and so should be prominent for clarity.

**Response:**
**We agree with the referee. This point should be clear as soon as possible in the manuscript. This point was already discussed in the originally submitted article (L461). We suggest adding the following words in the Abstract:**

**L4 of the originally submitted article:** "Here, by combining reanalysis data and climate model simulations, we investigate how two oceanic and two atmospheric variability modes -- ENSO, the Atlantic Multidecadal Variability (AMV), the North Atlantic Oscillation (NAO) and the Pacific North American (PNA) -- **amplify the wintertime (December–February) frequency of daily CWP extremes** and associated spatial co-occurrences across the Northern Hemisphere."

We also provide such information in the Introduction:

**L65 of the originally submitted article:** "**In this study, we use** reanalyses **data** and large ensemble climate model simulations from the CESM General Circulation Model (Kay et al., 2015)**, to investigate the effects of ENSO, NAO, PNA and AMV modes of variability, and their combinations, on the increase in the frequency of December–January–February daily CWP extremes across the Northern Hemisphere.** [...] Specifically, we (1) analyse  **how different** modes of variability  wintertime regional frequencies **(i.e., seasonal counts) of daily CWP extremes across individual regions** of  the Northern Hemisphere."

**Comments:**

Section 2.2.4 – This approach seems reasonable, and statistically significance testing is critical in a paper like this, although I'd need to do a really careful read in a revised manuscript. Illustratively, the permutation procedure would need to account for dependency / relationships between the modes, or statistical significance of any results could be over-estimated (i.e. appear significant when they are not). And, whether this has been done is not currently clear to me.

**Response:**

Thanks for your comment. According to our interpretation of the reviewer statement "the permutation procedure should take into account the dependency/relationship between modes", our approach accounts for such dependencies.

The permutation procedure is mainly used to test, for a given CWP metric, if the average of this metric under some combination of modes is different from that during neutral conditions. We specifically focused on what we called *direct* and *combined* effects of modes, that is, for each combination under study, we additionally condition the other modes in the neutral state (see L134-L145 of the article initially submitted).

To perform the test for a given combination (e.g., ENSO+NAO-), the values of the CWP metric is collected for 1) all seasons associated with ENSO+, NAO-, (AMV and PNA being neutral), and 2) all seasons associated with all modes being neutral. Then, the ratio of averages of the CWP metric collected for 1) and 2) is computed. The obtained ratio is the "observed value" of the test statistic. **The test statistic thus is derived from combinations modes that are directly derived from original data and that, as such, accounts for the dependency/relationships between modes under study.**

Then, in a second step, the values used for the calculations of the means are randomly reassigned without remplacement to 1) seasons with ENSO+, NAO-, AMV being neutral and PNA being neutral, and 2) seasons with neutral conditions. Corresponding averages and ratios are then computed. **The permutation procedure thus assesses the significance of the relationship between [a] CWP metrics and [b] the mode combination under study by randomly breaking the relationship. Note that the relationship between the modes is not broken here, but only the relationship between CWP and modes.**

**Comments:**
Results:

L211: biases w.r.t ERA5. Fair enough, although I expect any relevant ones to be explicitly referred back to and results interpreted in light of this during the Discussion.

**Response:**
Thanks for this comment. Although climate model biases were initially partly discussed in the Discussion section, we chose to better highlight what the relevant biases are and how some of our obtained results should be taken with caution in light of these biases.

In the Discussion section: "**However, several results lack direct support from existing literature. While they may represent novel findings, they should be interpreted cautiously, considering the biases of CESM simulations relative to ERA5. Notably, the direct effect of ENSO+ on CWP extreme frequencies in CESM simulations exhibits some inconsistencies when compared to ERA5 over Northern Africa (Fig. S3). Better understanding and confirming the influence of climate modes on arid regions (e.g., Northern Africa), where CWP events may be less intense than in other areas, can support adaptation and mitigation policies. While CWP extremes can serve as an important source of freshwater (e.g., Berdugo2020), they also present a significant flood risk (e.g., Yin2023)**."

**Comments:**
Section 3.1 & 3.2: From a reader's point of view, it would be nice if this were significantly shorter, drawing out the main points of interest (i.e. that are new). Please review Section 3.1 as in a number of places it starts to discuss / explain the results to a level that is at or above the limit expected in a Results section.

**Response:**
Thanks for this comment. We suggest shortening Section 3.1 and 3.2 (see the modified article for the provided modifications). We tried to focus on the main points of interest (new results), despite the fact that Section 3.2 has more new interesting points than Section 3.1 (as most of the results in Section 3.1 obtained can be matched with some existing papers). We also removed identified elements of discussion and explanations of the physical mechanisms from Section 3.1 and 3.2. In addition, we also identified elements of discussion on causal links in Section 3.3, that we removed and discussed later in the Discussion section.

**Comments:**
L257 – 'we move to discussing' Please do not move to discussing in the results section. Please discuss in the Discussion.

**Response:**
In addition to changing the content of the results section, we modify the specified sentence as follows:

L258 of the article originally submitted: "**In the following, we**  **focus on describing the effects of a selection of mode combinations and regions in Figs. 3-5. To maintain clarity and conciseness, we do not discuss all regions and mode combinations in the text, and readers can explore specific regional effects directly in the figures.**"

**Comments:**
L315-326 – This seems like an expansion of or repeat of Methods. Consider moving to methods.

**Response:**
Thanks for this comment. It is true that this paragraph can be considered as an expansion of Methods. However, we think that it allows a smooth transition between sections. In addition to L315-326, we also identified another mention of methodology in Results (L329-333):

L329: "In general, dependencies between counts of CWP extremes in different regions can favour such spatially compounding events (Bevacqua et al., 2021) because regions connected by positive dependencies tend to experience CWP extremes at the same time. Thus, as a first step in the investigation of spatially compounding events, we analyse such dependencies. This also provides preliminary information on groups of regions that may be affected by CWP extremes during the same winters."

To address the referee's concern, we suggest removing parts of these two elements from the Results section and combined this information in Methods starting at L117 of the article initially submitted:

L117: "As Metric 1 is derived for each region individually, the influence of variability modes on the high regional frequencies of CWP extremes across multiple regions in the same winter (i.e., spatially compounding events) cannot be deduced  **directly. For example, based on Metric 1, we find that the variability mode phase ENSO+ modulates regionally averaged CWP extreme frequencies for North America and Central Asia. However, a possibility could be that half of the winter seasons with ENSO+ leads to increased CWP extremes for North America only, while the other half affects Central Asia, thus not simultaneously. Examining the dependencies between counts of CWP extremes in different regions can provide preliminary information on spatially compounding events (e.g., Bevacqua et al., 2021) because regions connected by positive dependencies tend to experience CWP extremes at the same time. Thus, as a first step for investigating spatially compounding events, we analyse dependencies in Metric 1 computed for different regions, so as to provide preliminary information on groups of regions that may be affected by CWP extremes during the same winters.**

**Then, to examine spatially compounding events, we use two additional metrics. We employ these metrics to investigate the effects of variability modes on regional high frequencies of CWP extremes** across multiple regions in the same winter (Metric 2) and on the total population of the Northern Hemisphere exposed to CWP extremes in the same winter (Metric 3)."

We then make the transition between subsections 3.2 and 3.3 shorter, which helps with following the flow of the analyses while still reducing the text, following the referee suggestion:

L315: "Results from Fig. 3 and the summary in Fig. 4 cannot be used to conclude whether the effects of concurrent variability modes lead to spatially compounding CWP extremes, that is, **to** high wintertime frequencies of CWP extreme across multiple regions during the

same winter. These figures illustrate the effect of individual and concurrent variability modes on regionally averaged CWP extreme frequencies, which are derived for each region separately.  Nevertheless, the number of regions where each mode combination has significant effects in Fig. 4 (see numbers on the top of the matrix) suggests that some mode combinations may potentially lead to spatially compounding CWP extremes. For example, NAO-ENSO+ significantly enhances regional CWP extreme frequencies in eight regions, which means that if these regional effects of NAO-ENSO+ can manifest in the same winter, NAO-ENSO+ would lead to spatially compounding CWP extremes. **In the next section, we assess** whether individual and concurrent variability modes can lead to concurrent CWP extremes during the same winters across regions."

**Comments:**
L336-7 – This long-distance correlation is interesting. It is an example of the type of thing that could be expanded upon and discussed in a Discussion.

**Response:**
Thanks for this suggestion. We add in the Discussion section:
"Overall, although our aggregation in time and space may not be optimal for providing a fine-grained analysis of CWP events and additional modes may be relevant in some regions, this study provides a first comprehensive assessment of the interactions between multiple climate variability modes and the frequency of wintertime CWP extremes and associated spatially compounding events across regions of the Northern Hemisphere. **In particular, analyzing spatially compound events highlights the potential influence of variability modes that can link distant regions. These long-range relationships modulated by mode combinations can be explored in more detail, for example, with tailored experiments such as nudged atmospheric simulations."**

**Comments:**
L340 – 'we find that dependencies among regions overall enhance the potential for spatially compounding events' I am unsure how you can make this conclusion given that you were explicit earlier about only looking at enhancement not reduction of co-occurrence. Surely, both need to be looked at to comment on an overall effect.

**Response:**
Thanks for this comment. We think there is a misunderstanding here. Indeed, we refer to the fact that, in Fig. 5a, correlations between CWP extreme counts in the different regions are mainly positive. Regions that are positively correlated would tend to experience more CWP extremes at the same time than if CWP extremes in the regions were independent. Accordingly, Figure 5b shows that the dependency between the regions overall increases the number of regions that experience CWP extremes within the same winter, compared to the independent case. Thus, here the focus is on the increase of CWP extremes, compared to the independent case.

**Comments:**

L341 – This, and similar mentions of methodology in Results, should be put into Methods please.

**Response:**

Thanks. We removed the sentence and added it to the Methods. We did not find other mentions of methodology in Results that have not been already addressed.

L132 **"In addition to enabling quantifying the number of affected regions depending on variability modes, Metric 2 is also used in Fig. 5b to assess the effect of the dependencies between regions on spatially compounding events. This analysis is performed by comparing the number of affected regions (i.e., Metric 2) from the original dataset with the number obtained after breaking the dependencies via randomly shuffling regional CWP extreme counts using bootstrap in all regions in time (Bevacqua et al., 2021a)."**

**Comments:**
L366 – 'causal links among climate variability modes and oceanic modes exist'

**Response:**

It seems that there are no comments from the referee associated with this quote.

**Comments:**
Discussion:
L425-436 These are assertions, picking highlights from the results. These results are not discussed, i.e. reflected upon and put in the context of the literature. Suggest removing, or including in the Results.

**Response:**

Thanks for this comment. We moved and adapted it at the beginning of the Conclusion section.

**Comments:**
L437 – 445: Is a justification of the Methods, which I think is a repeat from the Methods section. Remove.

**Response:**

Thanks for this suggestion. We simply removed this part from the Conclusion section.

**Comments:**
L456 – This paragraph is a restatement of the approach, until L456 where an alignment with existing results is stated. So, it would be good to clarify what the new insights provided by this paper are.

**Response:**

Thanks for this comment. We suggest modifying (in blue) the following sentence starting L456 of the initial Manuscript to better clarify the new insights provided by our study.

L456: "Overall, although our aggregation in time and space may not be optimal for providing a fine-grained analysis of CWP events and additional modes may be relevant in some regions, **this study provides a first comprehensive assessment of the interactions between multiple climate variability modes and the frequency of wintertime CWP extremes and associated spatially compounding events across regions of the Northern Hemisphere**."

**Comments:**
L461 – This paragraph is a caveat, which is OK, but should come after a substantive discussion.

**Response:**
Thanks for this comment. We let this paragraph follow elements of the discussion, following the suggestion of the referee.

**Comments:**
Conclusions:
The conclusions are suitable in style, but are difficult to comment while the assertions being made have not previously been discussed.

**Response:**
Thanks for this comment. We tried to address this issue by rewriting the Discussion section.

**Comments:**
Fig. 6 – It's good to see the Bonferroni correction being used.

**Response:**
We thank the referee for the feedback.

---

## Author Comment (AC2)

**Response to Referee Comment 3: "Concurrent modes of climate variability linked to spatially compounding wind and precipitation extremes in the Northern Hemisphere"**

**Comments to the authors**

**Overview:**
This paper focuses on compound wind and precipitation (CWP) extremes, aiming to identify the drivers behind the occurrence of these events in the Northern Hemisphere. Climate model simulations from the Community Earth System Model are used with reanalysis data (ERA5) providing a "sense check". A few key climate variable modes are considered (ENSO, AMV, NAO & PNA). The individual effects of these events are found to follow existing literature, e.g. NAO+ increasing CWP extremes in Northern Europe. Concurrent phases of variability modes are considered with specific regional effects discussed. The NAO- & ENSO+ combination increased the likelihood of CWP extremes in eight regions. This motivated exploring spatially compounding extremes, where a positive trend between the number of anomalous variability modes and the number of regions was identified. Physical mechanisms for the statistical relationships were then discussed. This paper concludes ENSO is the most influential mode of variability for CWP extremes in the Northern Hemisphere.

Compound events are an area of current interest and this manuscript will appeal to the community. It is suitable for this NHESS special issue and I therefore recommend its publication subject to the changes outlined below. I would therefore appreciate the author's response on the comments below.

**Response:**
We would like to thank the referee for their positive comments and detailed feedback. All the comments and our point-by-point responses are given below.

**Comments:**

**General comments:**
As this study covers a large region and many combinations of variability modes, the presentation of results is important. The paper has a wide scope which at times means detail on specific regions is lacking. Choosing two or three regions or one teleconnection index to focus on gives this study more impact.

**Response:**
Thanks for this comment. We agree that the presentation of results is important, particularly for a broad study like ours that covers many regions and four variability modes and their combinations. The primary aim of this study is to explore the broad patterns of spatially compounding CWP extremes across the Northern Hemisphere. We understand the referee's point that choosing a reduced number of regions and teleconnection indexes would allow us to provide valuable insights for the specific modes and regions selected. However, we think that, by not choosing to focus on specific regions or teleconnection index, our study can provide more broad information. In particular, by providing the most significant results for multiple regions and combinations of modes, we envision the study providing a broad overview of the influence of combinations of modes on CWP extremes. In this direction, we

hope this overview will guide and motivate future, more targeted investigations into regions or modes of particular interest.

**Comments:**

While the standard of written English is fine, the language used makes this paper difficult to read at times. There are some very long sentences which could be split up or multiple sentences which may be more readable as a bullet pointed list.

**Response:**

Thanks for this comment. We recognize the importance of clear and concise language in scientific communication, particularly for a study with a broad scope and technical complexity. We worked on the paper to improve readability. While we aimed for a formal and detailed style, we split up long sentences to improve readability.

**Comments:**

Redrafting Section 3 will make the paper more readable and therefore accessible to the wider scientific community. Figures are meant to help convey information simply, Figures 3, 4 & 5 are complex. The authors should only include combinations of variability modes discussed in the text with the full figures available in the supplementary material.

**Response:**

We thank the reviewer for pointing this out. While accessibility and understandable figures are of great importance to us, we feel the message Figures 3, 4 and 5 convey is quite nuanced. To simplify them further, we would need to compromise on the clarity and preciseness of our message, which we feel would be detrimental for the paper. Although we agree simplified figures may more easily provide a first level of information to the reader, the current figures provide a crucial overview of the relationships between variability modes and CWP extremes and associated spatially compounding extremes, which is essential for understanding the global patterns we aim to highlight. Including all combinations allows readers to evaluate the relationships beyond the specific examples discussed in the text and, for example, distill those that are relevant to their region of interest. This broader context is also critical for readers interested in the full spectrum of variability mode interactions. For these arguments, we have decided to keep the figures in their current form. However, to ease their interpretation, we have improved and refined the text in Section 3 relating to the mentioned figures to better guide readers through the analyses, emphasizing the most critical patterns and combinations while leaving room for individual exploration of the full dataset. We also emphasized in the text that a selection of the various mode combinations is presented:

L258 of the article originally submitted: "**In the following, we focus on describing the effects of a selection of mode combinations and regions in Figs. 3-5. To maintain clarity and conciseness, we do not discuss all regions and mode combinations in the text, and readers can explore specific regional effects directly in the figures.**"

**Comments:**

The choices of percentile thresholds are arbitrary. The results of this study would hold more weight if a sensitivity analysis on these had been conducted. e.g. 98th percentile of daily precipitation seems low as this data is zero inflated.

**Response:**

Thanks for this comment. Although using different variables (that is, wind gusts instead of wind speed), some studies considered the local 98th percentile to investigate precipitation and wind extremes (Martius et al., 2016). Also, Klawa and Ulbrich (2003) show that the local 98th wind percentile is a damage-relevant wind threshold for wind gusts. In our study, daily data for the December-January-February months are used. It represents 90 days by season. By choosing the 98th percentile as a threshold, the expected number of exceedances per season for wind and precipitation in isolation is equal to 90*0.02 ≈ 2 events per season, which can be considered sufficient to analyse co-occurrences of wind and precipitation values above these thresholds. Choosing a percentile higher than the 98th would allow us to focus on more extreme events (e.g., Zhang et al., 2011), but it would reduce the sample size for the analyses. Note that for model evaluation, we use the 95th percentiles, to ensure a sufficiently large sample size given the shorter record length of reanalysis data. Although CWP events exceeding the 98th percentile of wind and precipitation can be considered moderate extremes, we think that they can still be considered impact-relevant. Choosing a local and impact-relevant threshold for wind and precipitation extremes is difficult, especially for precipitation for which incorporating effects such as surface runoff, snow melt and landslides would be needed (e.g., Williams, 1978) and is thus out of the scope of this study.

To expand on these trade-off issues between sufficiently large sample sizes and sufficiently extreme events, we suggest adding the following explanations (in blue) to the text:

**L110 of the article initially submitted:** "We use the 98th percentile of wind and precipitation over the 1950–2019 period for the main analysis based on data from the CESM model. **Percentile-based thresholds are frequently used to investigate climate extremes (e.g., Zhang et al., 2011, Martius et al., 2016). Following Klawa and Ulbrich (2003) and Martius et al (2016), we chose the 98th percentile, which is a compromise to capture the most extreme events in the CESM simulations while ensuring a sufficiently large sample size for robust statistical analysis.** For model evaluation**, which involves both the CESM model and ERA5 reanalyses (Figs. S1-S5 of the Supplement only)**, we use the 95th percentiles over the 1950-2019 period -- such a lower threshold allows for a more robust evaluation. **The reason for this is that,** given the ERA5's limited period, **extremes in the reanalysis data set are more scarce and associated statistics for very extreme events are largely affected by sampling uncertainty (Bevacqua et al., 2021b). Selecting a slightly lower threshold allows us to reduce this sampling uncertainty and thus improve confidence in assessing the model's ability to simulate extremes (e.g., Bevacqua et al., 2021b, Kelder et al., 2022, Fischer et al., 2023)**."

**References:**
- Zhang, X. B., L. Alexander, G. C. Hegerl, P. Jones, A. K. Tank, T. C. Peterson, B. Trewin, and F. W. Zwiers (2011), Indices for monitoring changes in extremes based on daily temperature and precipitation data, *Wires Clim. Change*, 2, 851–870.
- Klawa, M., and U. Ulbrich (2003), A model for the estimation of storm losses and the identification of severe winter storms in Germany, *Nat. Hazard Earth Syst. Sci*, 3, 725–732.

- Williams, G. P. (1978), Bank-full discharge of rivers, *Water Resour. Res.*, 14(6), 1141–1154, doi:10.1029/WR014i006p01141.

Following the suggestion of the referee, we produced a sensitivity analysis by considering the 99th and 99.5th percentile to define seasonal counts of CWP extremes. New Figures S16-S19 have been added to the Supplement and display the results we obtained for:

- Metric 1: the influence of individual and concurrent variability modes on regional wintertime CWP frequency (Figs. S16 and S17, same results as those presented in Fig. 3 but consider the 99th and the 99.5th percentile, respectively).

- Metrics 2 and 3: the influence of variability modes on spatially CWP extremes (Figs. S18 and S19, same results as those presented in Fig. 6 but consider the 99th and the 99.5th percentile, respectively).

Regarding Metric 1, by increasing the percentile, some differences can be observed (Figs. 3, S16, and S17). Although the effect of the combinations on CWP extremes remains generally consistent in magnitude across percentiles (not shown), increasing the threshold generally limits the test procedure to identifying significant combinations. Significant effects were detected in 20 regions when using the 98th percentile (Fig. 3). Increasing the percentile to the 99th (Fig. S16) and 99.5th (Fig. S17) led to detect significant effects in 17 and 11 of these regions, respectively. Such a systematic reduction in the number of regions when considering higher thresholds aligns with the fact that higher thresholds lead to more seasons without CWP events, making it more difficult to detect a significant signal for Metric 1. Still, the results obtained for the three different thresholds (98th, 99th, and 99.5th percentiles) are fairly consistent. In particular, (1) despite increasing the threshold generally limits the test procedure identifying all combinations that were significant at lower thresholds, the combinations detected at higher thresholds are consistently included among those identified at lower thresholds. Note that, in line with what was stated above, the differences might be due to increased sampling uncertainty associated with higher thresholds rather than differences in the involved physical mechanisms. Furthermore, (2) the magnitude of the effects of the combinations that were detected as significant at lower thresholds but not at higher thresholds are generally consistent across thresholds (not shown).

Regarding Metrics 2 and 3, we also observe some differences depending on the threshold (Figs. 6, S18 and S19). However, the main conclusions of our study are not changed: for the 99th and 99.5th percentiles, combinations of variability modes have a significant effect on the total number of affected regions (Figs. S18a and S19a), along with an amplified effect relative to their underlying mode sub-combinations, with ENSO+ being the predominant mode phase (see "+" sign). For the population affected, the influence of variability modes is primarily driven by ENSO- (see the '−' sign in Figs. S18b and S19b), consistent with the findings of the main study based on the 98th percentile (see Fig. 6b).

We added some sentences in the Discussion:

"We analysed event counts aggregated over winter and at the scale of predefined SREX regions, given that high counts of compound extremes at these scales are expected to have negative effects on society. **While the 98th percentile has been used in this study to**

**focus on extremes and is relatively well-established in the literature (e.g., Klawa et al., 2003, Martius et al., 2016), other higher thresholds could have been chosen to consider more intense extreme events (e.g., Liu et al., 2013, Schar et al., 2016, Camuffo et al., 2020). Figs. S16-S19 show results from a sensitivity analysis on the influence of variability modes on regional CWP extremes (Metric 1) and spatially compounding events (Metrics 2 and 3) with the 99th and 99.5th percentiles used as thresholds. Although there are some variations in the results compared to those for the 98th percentile, the main conclusions drawn across the different thresholds are broadly consistent for all Metrics. The magnitude of the effects of the combinations are generally consistent across thresholds, and the combinations detected at higher thresholds are generally included among those identified at lower thresholds (Figs. S16-S19). Such slight differences may be due to larger sampling uncertainty for higher thresholds limiting the ability to detect significant effects for higher thresholds rather than different physical mechanisms involved for different thresholds. While the sensitivity analyses broadly indicate the robustness of most of our findings, possible relevant differences across thresholds highlight the importance of identifying impact-relevant thresholds, though this task is challenging (Williams, 1978, Bloomfield et al., 2023).** In addition, the selected SREX regions may not reflect the natural spatial patterns of variation of CWP extremes, potentially occurring at a more localized scale or span across multiple regions."

**Comments:**

Daily precipitation is not always proportional to any resulting impact – the authors should acknowledge the complexity of the precipitation-flood relationship. For more on this see Bloomfield et al. (2023) [ https://doi.org/10.1016/j.wace.2023.100550 ]. While compound wind-precipitation events cause large impacts, they are rare (e.g. Fig. 2 from Jones et al. (2024) [ https://doi.org/10.1002/wea.4573 ] ). Considering these extremes in isolation gives the complete picture of a compound hazard.

**Response:**

Thanks for this comment. We agree that the relationship between daily precipitation and its impacts, particularly flooding, is highly complex and not necessarily proportional. This complexity arises from numerous factors, such as antecedent soil moisture conditions, land use, and drainage capacity, which influence the translation of precipitation into flooding. We will acknowledge this in the Discussion section and cite Bloomfield et al. (2023) to provide additional context.

Regarding the rarity of compound wind-precipitation events, as noted by Jones et al. (2024), we agree that their infrequent nature does not diminish their potential for significant societal and environmental impacts. Our focus on these events aims to understand the drivers and spatial relationships of compound wind-precipitation extremes. While considering precipitation and wind extremes in isolation may offer valuable insights, our study aims to focus on compound events explicitly.

We added some sentences in the Discussion:
"**While the sensitivity analyses broadly indicate the robustness of most of our findings, possible relevant differences across thresholds highlight the importance of**

**identifying impact-relevant thresholds, though this task is challenging (Williams, 1978, Bloomfield et al., 2023).**"

**Comments:**
You have cited Manning et al. (2024) to highlight extratropical cyclones as drivers of CWP events, but Manning et al. (2024) notes CWP events can be driven by precipitation extremes.

**Response:**
Thanks for this comment. Indeed, Manning states that the expected increase in precipitation due to the influence of climate change will make compound wind and precipitation extremes more likely, and that they will be produced by extratropical cyclones. We really don't see a contradiction here, but we understand the importance of your statement. This paper (from Owen et al., https://www.sciencedirect.com/science/article/pii/S2212094721000384#sec4) is more adequate to support our point, therefore we added this reference to support the statement.

**Comments:**
Specific comments:
L2: Change "agricultural crops" to "crops"

**Response:**
Thanks for this comment. We changed the text accordingly.

**Comments:**
L6: Remove NAO & PNA abbreviations, they are not used in rest of abstract.

**Response:**
Thanks for this comment. However, for consistency, we kept all the abbreviations in the Abstract, which are then used in the rest of the study.

**Comments:**
L13: Remove "For example" here, the reader knows you're giving them an example.

**Response:**
Thanks for this comment. However, we think that "for example" is important in the structure of the sentence.

**Comments:**
L17-22: Split into two sentences and rejig. Define compound events first, then highlight their importance from this IPCC report.

**Response:**
Thanks for this comment. We split the sentence into two as follows: "**Compound weather and climate events, defined as the combination of multiple drivers and/or hazards that contribute to societal or environmental risk, often cause more severe impacts than the respective single hazards (Zscheischler et al., 2018). The Intergovernmental Panel on Climate Change (IPCC) Special Report on Managing the Risks of Extreme Events**

**and Disasters to Advance Climate Change Adaptation (SREX) highlighted the importance of studying compound events to improve modeling and risk estimation of weather impacts (IPCC, 2012)**".

**Comments:**
L51: Useful to describe what the deviation from mean NAO conditions is, how does it affect frequency & intensity of events?

**Response:**
Thanks for this comment. We changed the text (in blue) as follows:

L51: "During extreme phases of the PNA and NAO, the intensity and location of storms and moisture transport deviate from mean conditions over the Pacific-North American region (e.g., Wallace et al, 1981, Xie et al., 2020) and the Euro-Atlantic region (e.g., Hurrell et al., 2003, Lodise et al., 2022), respectively. **While positive NAO phases intensify westerly winds and shift the North Atlantic storm track toward the northeast, leading to increased storm frequency and intensity over Northern Europe, negative NAO phases weaken the westerlies and amplify storm activity in the Mediterranean region (e.g., Hurrell and Deser, 2010)."**

**Comments:**
L74: Specify which months the winter season covers.

**Response:**
Thanks for this comment. We now precise in the Abstract and Introduction which months the winter season covers in our study.

**Comments:**
L75: Change "effective" to "influential"

**Response:**
This sentence is not part of the Manuscript after incorporating the changes from the other reviewers.

**Comments:**
L84: Make the rationale behind the choice of these regions clearer. These shapes cut across country boundaries, making this study less applicable to the insurance industry.

**Response:**
Thanks for this comment. We modified the text as it follows (in blue):

"We examine the influence of four variability modes on CWP extremes across 25 selected regions in the Northern Hemisphere defined in the SREX (Iturbide et al., 2020, see Fig. 1). **We chose these regions as they are standard reference in IPCC reports, as they encompass areas with relatively homogeneous climatic characteristics (Iturbide et al., 2020). While using these regions does not enable an explicit analysis of dependencies between local-scale CWP extremes and modes of variability, it allows for complementing IPCC assessments.**"

**Comments:**
L96: Why did you choose to begin with 1959? ERA5 covers from 1940 so matching the same period as CESM makes sense.

**Response:**
Thanks for this comment. When we started the project in 2022, ERA5 1940-1958 was not available. As a result, it was not possible to incorporate this data during the first steps of our project. While ERA5 data for 1940-1958 is now available, its quality and reliability for this period remain questionable due to sparse observational input, as acknowledged by the Copernicus Climate Change Service (https://www.ecmwf.int/en/newsletter/175/news/era5-reanalysis-now-available-1940). For this reason, we opt not to use it, as it would potentially compromise the robustness of our analysis and it would require an important computation effort.

**Comments:**
L96: "Singh et al. (2021)" reference doesn't make sense here? As far as I can tell, Singh et al. (2021) doesn't use ERA5?

**Response:**
Thanks for this comment. We removed the reference.

**Comments:**
L110: The 95th percentile of daily data considers 1114 days in this period (1959-2019) to be extreme. Yet the 98th percentile over 1950-2019 only considers 511 extreme days. Surely a higher threshold of ERA5 data is required for these periods to be comparable?

**Response:**
We want to thank the reviewer for this comment, as some clarifications are needed. In the study, we consider two percentile-based thresholds to determine CWP extremes: 98th percentile of wind and precipitation for the main analysis (for CESM) to ensure a sufficiently large sample size of simulated events while assessing extremes sufficiently extreme, and the 95th percentile for model evaluation only (both for CESM and ERA5; Figs. S1-S5 of the Supplement) to ensure a sufficiently large sample size in the reanalysis data which is only one realization over a shorter record. Therefore, thresholds are identical when comparing occurrences of CWP extremes in CESM simulations and ERA5 data. We suggest to provide the following clarifications (in blue) to the text:

**L110 of the article initially submitted:** "We use the 98th percentile of wind and precipitation over the 1950–2019 period for the main analysis based on data from the CESM model. **Percentile-based thresholds are frequently used to investigate climate extremes (e.g., Zhang et al., 2011, Martius et al., 2016). Following Klawa and Ulbrich (2003) and Martius et al (2016), we chose the 98th percentile, which is a compromise to capture the most extreme events in the CESM simulations while ensuring a sufficiently large sample size for robust statistical analysis.** For model evaluation**, which involves both the CESM model and ERA5 reanalyses (Figs. S1-S5 of the Supplement only)**, we use the 95th percentiles over the 1950-2019 period -- such a lower threshold allows for a more robust evaluation. **The reason for this is that,** given the ERA5's

limited period, **extremes in the reanalysis data set are more scarce and associated statistics for very extreme events are largely affected by sampling uncertainty (Bevacqua et al., 2021b). Selecting a slightly lower threshold allows us to reduce this sampling uncertainty and thus improve confidence in assessing the model's ability to simulate extremes (e.g., Bevacqua et al., 2021b, Kelder et al., 2022, Fischer et al., 2023)**."

**Comments:**
L115: Include rationale for weighting by cosine of latitude.

**Response:**
Thanks for this comment. The weighting by the cosine of latitude is applied to account for the spherical geometry of the Earth. Without this correction, grid cells closer to the poles, which cover smaller physical areas, would be overrepresented in the analysis compared to those near the equator, which cover larger areas. This approach ensures that regional averages are spatially representative, reflecting the actual physical extent of each grid cell.

By applying this weighting, we maintain consistency with standard practices in climate and atmospheric sciences, ensuring that the metrics derived are not artificially biased by the unequal spatial resolution inherent to a latitude-longitude grid system. This correction is particularly important in studies like ours that involve large-scale regional analyses across diverse latitudes.

We included the rationale as follows:

L115: Wintertime CWP counts are averaged by region over landmasses, weighted by the cosine of latitude **to prevent overrepresentation of grid cells closer to the poles.**

**Comments:**
L151: Change "That is, in this study, we do not…" to "This study does not".

**Response:**
Thanks for this comment. We changed the text accordingly.

**Comments:**
L154: Remove ", in principle,"

**Response:**
We changed the text accordingly.

**Comments:**
L162: The 280 year return period seems to be an arbitrary choice. Sensitivity analysis on this threshold would be of interest.

**Response:**
Thanks for this comment. The choice on the return period is required to ensure that the combinations of variability modes analyzed had a sufficiently large sample size for robust statistical assessment while focusing on relatively rare, impactful events. It was decided to

consider samples of minimum size 10 years, which given the yearly resolution of the aggregated data implies to consider combinations occurring more than 10 years in our 2800-year dataset. Given the length of our 2800-year dataset, this implies exploring mode combinations with a maximum return period of 280 years. Changing a different return period as a threshold would only change the combinations of modes displayed in the figure, but importantly, the effects presented for the combinations illustrated in the submitted paper would not change.

**Comments:**
L176: Mismatched bracket after "subsection 2.2.3".

**Response:**
Thanks for this comment. The mismatched bracket was deleted.

**Comments:**
L180: A 10% significance level seems high, 5% (or even 1%) level is much more standard practice.

**Response:**
Thanks for this comment. The choice of a 10% significance level was intentional to balance the detection of meaningful effects while avoiding false negatives. With a limited sample size (which is the case when we compare distributions in the study), a lower threshold might be too stringent to detect meaningful effects, especially at a 5 or 1% significance level. Choosing a larger significance level is aligned with the exploratory nature of our work, allowing us to shed light on potential effects of modes on CWP extremes.

We suggest to add the following sentence:
**L180 of the article initially submitted:** "Specifically, for a given CWP metric, we test whether the ratio of the average of the metric associated with a given set of phases of interest (e.g., NAO+ENSO-, set as the numerator) to the average of the metric under neutral conditions (set a denominator) is larger than one at significance level α = 0.10 based on one-sided tests. **Compared to a lower significance level, our chosen level allows the detection of significant effects of modes of variability while reducing false negatives in the context of small sample sizes.**"

**Comments:**
L185: How many times is "several times"? State this in the text.

**Response:**
Thanks for this comment. Here "several times" is used to describe the concept behind permutation testing, therefore we did not deem it necessary to give the exact number of permutations. Depending on the case, we used a different number of permutations: (m=100,000 for the analysis of the three metrics; m=100 when applied to the grid cell level for Figs. 2 and 7). We will change the text in this way to avoid the confusion.

L185: "By repeating this procedure , we can then define a confidence interval for the ratio and a critical region for test rejection."

**Comments:**

L199-200: Change 100.000 to 100,000

**Response:**

Thanks for this comment. We changed the text accordingly.

**Comments:**

L224: A significant body of literature exists linking extreme windstorms to strong winds (favourable conditions for CWP events). Here I would at least cite:
- Mailier et al. (2006) https://doi.org/10.1175/MWR3160.1
- Priestley et al. (2024): https://doi.org/10.5194/nhess-23-3845-2023

**Response:**

Thanks for this comment. We no longer discuss the findings in the Results section. References are thus not needed here. The reference for Priestley et al., 2024 has been however added to the Introduction.

**Comments:**

L312: I'd make this sentence clearer, "generally covers most of the time" is very ambiguous.

**Response:**

We agree with the reviewer. We suggest deleting "most of the time" in the text.

**Comments:**

L391: Change "Europa" to "Europe".

**Response:**

Thanks for this comment. We changed it.

**Comments:**

L430: Is this not driven by atmospheric circulation patterns?

**Response:**

Thanks for this comment. While we agree with the referee, this sentence has been removed from the text.

**Comments:**

L480: Change "found" to "estimated"

**Response:**

Thanks for this comment. We suggest changing the text as follows (in blue):

L479-480: "By repeating this procedure among different modes, we  estimated a wide range of return periods for the different mode combinations (Fig. 3)."

**Comments:**

L484: A natural next step would be repeating this study for the southern hemisphere.

**Response:**

Thanks for this comment. The current study focuses on the Northern Hemisphere due to its dense population and economic significance. Indeed, we acknowledge that extending this study to the Southern Hemisphere would be a valuable next step, considering the relevant variability modes for that Hemisphere and meteorological season. We now mention it in the Discussion.

---

## Author Comment (AC3)

**Response to Referee Comment 2: "Concurrent modes of climate variability linked to spatially compounding wind and precipitation extremes in the Northern Hemisphere"**

**Comments to the authors**

**Overall impression of article**
I think it is an important study on compounding extreme events, especially when linking this to potential impact (e.g. through the population metric used in this study). I also value the link to global drivers and teleconnections, as this can benefit short term predictions but also long term projections. Generally, I think this paper needs a bit more restructuring, notably the result and discussion sections. Furthermore, it needs another careful read through because it was difficult at times to understand the sentences. Below I mention more details in some major and some minor suggestions.

**Response:**
We would like to thank the referee for their positive comments and the detailed feedback. We took action and rephrased sentences that were too long or complex when needed. All the comments and our point-by-point responses are given below.

**Comments:**
**Major points of discussion**
1. The motivation for this study is not so clear to me. Why look at compound wind&precip?

**Response:**
Thanks for this comment. In this study, we looked at compound wind and precipitation (CWP) extremes and associated spatially compounding events due to the potential widespread impacts it could have on many sectors of the economy (such as damages to agricultural crops, energy infrastructure and buildings). As suggested by the referee later in the review, CWP events are not the only impactful compound events. Droughts and heatwaves or floods are indeed other examples of potentially impactful spatially compounding events that need to be taken into account for risk assessments. However, to our knowledge, spatially compounding wind and precipitation extremes and their drivers have been little investigated, representing a research gap in the literature.

We add in the Abstract some sentences (in blue) to clarify the motivation of the study:

**L1:** "Compound wind and precipitation (CWP) extremes often cause severe impacts on human society and ecosystems, such as damage to crops and infrastructure.  **Spatially compounding events with multiple regions affected by CWP extremes in the same winter can impact the global economy and reinsurance industry, however our understanding of these events is limited. While climate variability modes such as El Niño Southern Oscillation (ENSO) can influence the frequency of precipitation and wind extremes, their individual and combined effects on spatial co-occurrences of CWP extremes across the Northern Hemisphere have not been systematically examined.**"

Link to this comment, the referee later questions why CWP should be prioritized over other multi-hazard events. Therefore, we also highlight other significant compound events in the Introduction:

**L24 of the article initially submitted:** "**A highly studied example of an impactful compound event is hot-dry conditions (e.g., Bevacqua et al., 2022), which are enhanced by land-atmopshere feedback (e.g., Zscheischler et al., 2017, Rasmijn et al., 2018, Ridder et al., 2022). Another example is compound wind and precipitation (CWP) extremes that can cause more damage than high winds or precipitation in isolation (e.g., Li et al., 2024)**."

We better motivate the reason why we specifically analyse CWP extremes in the Introduction (instead of another compound events):

**L40 of the article initially submitted:** "**While the characteristics of spatially compounding events such as droughts (e.g., Kim et al., 2019, Singh et al., 2021) or floods (e.g., Jongman et al., 2014) have already been studied, our understanding of spatially compounding CWP extremes and their drivers is limited.**"

**Comments:**
2. Similarly, the motivation for these exact modes of variability is a bit lacking in introduction. You do mention the Indian ocean as a potential influence in discussion. What about other modes? Why not include those?

**Response:**
Thanks for this comment. The four climate modes NAO, PNA, AMV and ENSO are known to strongly influence wintertime storm activity, atmospheric circulation, and moisture transport across the Northern Hemisphere, and are thus chosen in this study to investigate their relationships with occurrences of wintertime CWP extremes over the Northern Hemisphere. In the Introduction, we modified the examples of the already-known influence of climate modes on wintertime weather to specifically focus on the Northern Hemisphere.

**L47 of the article initially submitted:** "For example, the **North Atlantic Oscillation (NAO)** and Pacific North American (PNA)  **are the leading modes of variability affecting wintertime weather in Europe (e.g., Hurrell et al., 1995) and North America (e.g., Wallace et al., 1981).**"

**L56 of the article initially submitted:** "The AMV, an alternation of warm and cold sea surface temperatures in the North Atlantic on decadal timescales, has been shown to influence  **weather in both North America and Europe (e.g., Knight et al., 2006), as well as** the long-term variability of the NAO (e.g., Davini et al., 2015)."

We also specify precisely that the four modes are selected due to their already-known influence on CWP extremes in the Northern Hemisphere:

**L67 of the article initially submitted:** "We consider these four modes of variability due to their already-known influence on storm activity and moisture transport in the Northern Hemisphere."

We recognize that adding other climate modes, such as the Indian Ocean Dipole, that are also known to have an influence on wind and precipitation extremes would have been interesting. However, a choice on the number of modes has to be made, and here we selected four modes of variability that are known for having large effects on storm activity in winter across the Northern Hemisphere. We also note the addition of extra conditioning would result in a reduced sample size when quantifying conditional statistics, potentially reducing the robustness of the results of some statistical analyses, particularly those based on reanalysis data.

**Comments:**
3. Why do you choose to average the daily wind and precipitation values instead of taking the maximum wind speed and the sum of total precipitation of the day? Especially when it comes to wind, I'm worried that averaging is not the best choice to catch wind-extreme events.

**Response:**
We thank the referee as clarifications and corrections in the Manuscript are indeed required. First, regarding precipitation, we used daily data of total precipitation and not daily means, as indicated in the article. It should be noted, however, that as our methodology uses a quantile-based approach, the choice of daily average or daily total for precipitation does not affect our analysis by construction. Secondly, with regard to the wind values, the reviewer is right, and it would have been preferable to take into account the maximum wind speed rather than the daily averages to examine the wind extremes. This choice has been made due to data availability: for the CESM simulations, only daily averages were available.

We suggest to provide the following corrections to the text:

**L91 of the article initially submitted:** " **Daily total precipitation and daily mean of wind speed are extracted for** the historical period  (1950–2005). **Simulated data are then** extended until 2019 using the emission scenario associated with a radiative forcing of +8.5W.m−2 (RCP 8.5 scenario), resulting in a total of 70 × 40 = 2800 years of data. **We choose daily averages rather than maxima for wind due to data availability for the CESM simulations.**"

For ERA5, we directly downloaded daily means of wind speed and daily total precipitation, and did not get daily means from sub-daily data, as initially indicated in the article. Daily means were selected for consistency with the CESM data. We suggest the following correction:

**L95 of the article initially submitted:** "To evaluate the CESM model, we employ ERA5 reanalysis data (Hersbach et al., 2020) (spatial resolution of 0.25 ◦ ) for the period 1959-2019 (Singh et al., 2021), from which we  **also extract** daily means of wind speed and **daily total** precipitation **for consistency**."

**Comments:**
4. Why are you considering seasonal mean indices? Why not look at weekly/monthly data? I think there is an issue with the different timelags here, since ENSO is clearly a yearly oscillation, but the NAO can also be defined on weekly/monthly timescales. I think this has to be motivated from a physical point of view.

**Response:**
We thank the reviewer for raising this important point regarding the choice of seasonal mean indices for modes of variability. The choice of the seasonal time scale was made based on the modes considered in the study. The NAO has variability that differs in a physical way depending on the timescale (e.g. between interannual/decadal and multidecadal time scales, see e.g. Woollings et al., 2014, https://link.springer.com/article/10.1007/s00382-014-2237-y). As noted by the reviewer, ENSO operates primarily on seasonal timescales (e.g., Schmidt et al., 2001; Camberlin et al., 2001). Given one of the main focus areas of this study is on ENSO, it makes sense to consider this timescale in our study. Using the same timescale for all indices allows for a consistent comparison of their influences on CWP extremes. Considering shorter timescales (e.g., weekly or monthly indices) would have not allowed us to assess the influence of important modes such as ENSO and its interactions with other variability modes occurring at seasonal timescales.

We suggest to add in the Introduction:
"**We consider these four modes of variability due to their already-known influence on storm activity and moisture transport in the Northern Hemisphere. As ENSO operates primarily on seasonal timescales (e.g., Schmidt et al., 2001; Camberlin et al., 2001), seasonal mean indices are considered.**"

References:

- Schmidt, N., E. K. Lipp, J. B. Rose, and M. E. Luther, 2001: ENSO Influences on Seasonal Rainfall and River Discharge in Florida. *J. Climate*, **14**, 615–628.
- Camberlin, P., Janicot, S. and Poccard, I. (2001), Seasonality and atmospheric dynamics of the teleconnection between African rainfall and tropical sea-surface temperature: Atlantic vs. ENSO. Int. J. Climatol., 21: 973-1005. https://doi.org/10.1002/joc.673

**Comments:**
5. Which threshold do you end up choosing? It is a bit unclear, you take 95th in ERA5 and 98th in CESM? How do these two compare to each other (I believe you compare 95th in both era and CESM in the supplementary)? Did you do sensitivity experiments to determine these two thresholds are the same?

**Response:**
We thank the reviewer for this comment, as some clarifications are needed. In the study, we consider two percentile-based thresholds to determine CWP extremes: (1) 98th percentile of wind and precipitation for the main analysis based on the CESM model, and (2) the 95th percentile for model evaluation based on ERA5 and CESM model (Figs. S1-S5 of the Supplement). With respect to point 2, thus, contrary to what is initially understood by the reviewer, we use identical thresholds to compare occurrences of CWP extremes in CESM

simulations and ERA5 data. Therefore, no sensitivity experiments are required to determine if these thresholds are the same. We suggest to provide the following clarifications (in blue) to the text:

**L110 of the article initially submitted:** "We use the 98th percentile of wind and precipitation over the 1950–2019 period for the main analysis based on data from the CESM model. **Percentile-based thresholds are frequently used to investigate climate extremes (e.g., Zhang et al., 2011, Martius et al., 2016). Following Klawa and Ulbrich (2003) and Martius et al (2016), we chose the 98th percentile, which is a compromise to capture the most extreme events in the CESM simulations while ensuring a sufficiently large sample size for robust statistical analysis.** For model evaluation**, which involves both the CESM model and ERA5 reanalyses (Figs. S1-S5 of the Supplement only)**, we use the 95th percentiles over the 1950-2019 period -- such a lower threshold allows for a more robust evaluation. **The reason for this is that,** given the ERA5's limited period, **extremes in the reanalysis data set are more scarce and associated statistics for very extreme events are largely affected by sampling uncertainty (Bevacqua et al., 2021b). Selecting a slightly lower threshold allows us to reduce this sampling uncertainty and thus improve confidence in assessing the model's ability to simulate extremes (e.g., Bevacqua et al., 2021b, Kelder et al., 2022, Fischer et al., 2023)**."

**Comments:**
6. I think the result & discussion sections should be re-structured: you already discuss the findings with respect to other literature in the results, I believe this should be moved to the discussion. In the results only mention your own findings. This will also make your paper easier to read.

**Response:**
We agree with this comment. We removed aspects related to the discussion of the findings from the Results section to the Discussion.

**Comments:**
**Minor suggestions**
Abstract
1. I miss the motivation for these specific SST-modes of variability in the abstract.

**Response:**
As previously mentioned, the motivation for these specific SST modes of variability is now provided in the Introduction. To maintain a concise Abstract, we have opted not to include this motivation there.

**Comments:**
2. In the abstract I had to read the following sentence a few times before I understood: "we identify dependencies enabling extreme spatially compounding events with many regions experiencing CWP extremes in the same winter" L9/10.

**Response:**
Thanks for this comment. We simplified the sentence as:

**L9/10:** "By examining the relationships between **frequencies of wintertime** CWP extremes across regions, we identify dependencies enabling extreme spatially compounding events **, that is winters with many regions experiencing CWP extremes.**"

**Comments:**
3. "mitigation of spatially compounding CWP extremes." L15 how could these CWP extremes be mitigated ?

**Response:**
We suggest to modify the sentence as follows:
**L15:** "Our analysis highlights the importance of considering the interplay between variability modes to improve risk management and **adapt to the impacts of** spatially compounding CWP extremes."

**Comments:**
Introduction
4. L24: "co-occurring compound wind and precipitation (CWP) extremes" co-occurring and compound is that not the same?

**Response:**
Thanks for noticing this. We removed "co-occurring".

**Comments:**
5. Introduction: are there any examples of spatially compounding CWP events that lead to extreme damages? You mention the flooding as an example. But it is not entirely clear to me why CWP should specifically be investigated over other multi-hazard events (hot-dry, no wind-cold, etc.)

**Response:**
Thanks for this comment. Examining the drivers of spatially compounding of CWP events addresses a gap in the existing literature, compared to other compound events such as floods or droughts, which have been already investigated.

**Comments:**
6. L46: "cyclones are particularly exposed to CWP extremes" are cyclones not considered a CWP extreme? How is a CWP defined actually?

**Response:**
The referee refers to the sentence "regions prone to cyclones are particularly exposed to CWP extremes". Following the proposed typology for compound events (see Table 1 in Zscheischler et al., 2020), cyclones are considered a driver of CWP extremes. Thus, cyclones are not a CWP extreme. Furthermore, not all CWPs arise from cyclones, as explained in the Introduction (L43 of the article initially submitted). For instance, CWPs can occur in non-cyclonic systems, such as atmospheric rivers or frontal systems.

Reference:

- Zscheischler, Jakob & Martius, Olivia & Westra, Seth & Bevacqua, Emanuele & Raymond, Colin & Horton, Radley & Hurk, Bart & AghaKouchak, Amir & Jézéquel, Aglaé & Mahecha, Miguel & Maraun, Douglas & Ramos, Alexandre & Ridder, Nina & Thiery, Wim & Vignotto, Edoardo. (2020). A typology of compound weather and climate events. Nature Reviews Earth & Environment. 1. 1-15. 10.1038/s43017-020-0060-z.

**Comments:**
7. Why do you focus on wintertime CWP only? Aren't summer storms especially damaging (due to trees being in full leaves).

**Response:**
Thanks for raising this point. The occurrence of CWP extremes are known to be most frequent in the winter season, hence the choice of this season for our study.

We add the following sentence in the Introduction:
**"Such compound extremes are most frequent in coastal regions of the Pacific and Atlantic oceans (e.g., Maraun et al., 2016) and tend to be more frequent and intense in the winter season (e.g., Greeves et al., 2007, Hansen et al., 2019)."**

Still, we acknowledge that summer storms can be particularly damaging due to trees being in full leaf, and thus, it is interesting to analyse. Future research could address these differences and explore the vulnerabilities associated with summer storms. For example, summer storms are known to be more localized (e.g., convective storms) than winter storms, potentially resulting in different conclusions. We now also mention that summer storms are interesting to analyse in the discussion section.

**Comments:**
8. L156 how do you calculate significance?

**Response:**
Thanks for this comment. All the details are provided in subsection 2.2.4, as already specified. We suggest modifying the following sentence (in blue) to point to the text explaining the calculation of significance:

L156 of the article initially submitted: "When presenting the results in Sect. 3, we focus our analysis on individual and concurrent variability modes having a *significant and positive* effect on the different metrics using permutation tests (see subsection 2.2.4 **for more details on calculating the significance**)."

**Comments:**
Methods
9. Metric 1: if you average the count per grid point do you still need the latitude weight?

**Response:**
Thanks for this question. As indicated in the text for Metric 1 of the original manuscript, "CWP counts are averaged by region over land-masses, weighted by the cosine of latitude". The reason for using latitude weighting when averaging over regions is to account for the

potential uneven distribution of grid points across latitudes in the different regions. Without weighting, grid cells closer to the poles would be disproportionately represented in our calculations, potentially distorting regional averages for regions including such grid cells, for example Greenland/Iceland or Northeastern North America. Thus, we still need the latitude weighting for Metric 1.

**Comments:**
10. Metric 2: why 80th percentile?

**Response:**
Thanks for this comment. Choosing a higher threshold would allow us to focus on regions that are more severely affected. However, it would also limit the number of regions being reported as affected and, thus, the statistical robustness of our results regarding spatially compounding extremes. We suggest to add the following sentence:

L125 of the article initially submitted: **"Although choosing a higher percentile (>80th percentile) would enable us to focus on cases where regions are more severely affected, it would considerably limit the number of regions reported as affected and, consequently, the statistical robustness of our results."**

**Comments:**
11. Why do you only look at positive cases, e.g. when a mode has a positive effect? L151

**Response:**
We decided to look at positive cases, i.e., when anomalous phases of variability modes lead to an increase in the means of the different metrics compared to neutral conditions because of the greater potential impact of these events, and therefore their societal relevance. Indeed, looking at positive cases means looking at cases for which modes increase the frequency of CWP extremes (Metric 1) or the number of regions affected by CWP extremes (Metric 2). We added the following words to the text:

**L151 of the article initially submitted:** "For both regional and spatially compounding cases, we focus on the modes of variability influences leading to an **enhancement** in the means of the different metrics compared to neutral conditions (hereafter referred to as *positive effects*) **– we focus on the increase, rather than the decrease, as this is of relevance for potential impact to society.**"

**Comments:**
12. Have you tried any kind of regression analysis? Maybe this also can take away the effect of 'neutral' states not really being neutral, as mentioned L139-140

**Response:**
Thanks for this comment. It is a very good point from the reviewer, as using regression techniques could help better controlling the effect of modes varying within the range of neutral conditions. We did not try any kind of regression analysis and represents a nice perspective for future research. We modified a sentence in the Discussion section to mention it as a possible perspective for future studies.

**Comments:**
13. Why not take the significance level of 0.05? L 180: significance level α = 0.10

**Response:**
Thanks for this comment. The choice of the significance level is always subjective. Here, the 10% significance level was intentionally chosen to balance the detection of meaningful effects while avoiding false negatives. Indeed, with a limited sample size (which is the case in our study when we compare distributions), a lower significance level might be too stringent to detect significant effects, especially at a 5% or 1% significance level. Choosing a larger significance level is aligned with the exploratory nature of our work, allowing us to shed light on potential effects of modes on CWP extremes.

We suggest to add the following sentence:
**L180 of the article initially submitted:** "Specifically, for a given CWP metric, we test whether the ratio of the average of the metric associated with a given set of phases of interest (e.g., NAO+ENSO-, set as the numerator) to the average of the metric under neutral conditions (set a denominator) is larger than one at significance level α = 0.10 based on one-sided tests. **Compared to a lower significance level, our chosen level allows the detection of significant effects of modes of variability while reducing false negatives in the context of small sample sizes.**"

**Comments:**
Results

14. Your maps would be easier to interpret if you mask out the non-land areas.

**Response:**
Thanks for your comment. Although we agree that such masking would allow a more direct focus on regional CWP extremes, it is interest to some readers to have an overview of the occurrences of CWP extremes over the oceans, which also allows for having spatial continuation in the visualisation of the effects across the earth surface and facilitate interpretation. Consequently, we decided not to mask the non-land areas as suggested by the referee and let the maps be as they are.

**Comments:**
15. What's the difference between the following two statements in section 3.2 L250 and L259: "Model simulations (CESM) show that not only individual variability modes can have effects on regional wintertime frequencies of CWP extremes, but also combinations of modes." vs "Model simulations (CESM) show that concurrent anomalies in variability modes amplify the effects of individual modes in many regions." I think this section needs more attention. There are so many details in the figure, and the text is not complimenting this enough. It is very difficult to understand the main results at the moment, also because you weave discussion in here.

**Response:**
We want to thank the referee for this comment. First, we removed L259 as it is redundant with L250, as identified by the referee. Second, we simplified Section 3.2 and reduced its size by focusing on the main results. We also removed the elements of discussion. As noted

by the reviewer, a lot of information is provided by Figures 3 and 4, and is not possible to fully describe in the text. We suggest to better precise this point in the following sentence:

L258 of the article originally submitted: "**In the following, we**  **focus on describing the effects of a selection of mode combinations and regions in Figs. 3-5. To maintain clarity and conciseness, we do not discuss all regions and mode combinations in the text, and readers can explore specific regional effects directly in the figures.**"

**Comments:**
16. L118-119: "in general agreement with existing literature,"; either mention the literature or do not mention this. Generally, I think this should be part of the discussion, not the results.

**Response:**
Thanks for this comment. We removed all the mentions to the existing literature from Section 3.1 and 3.2 and instead mentioned it in the Discussion section.

**Comments:**
17. L315-325 suits better in discussion?

**Response:**
We believe the referee wanted to say "Methods" instead of "Discussion". We moved and adapted these lines to the Methods section, in agreement with another comment from the first reviewer.

**L116 of the article initially submitted:** "As Metric 1 is derived for each region individually, the influence of variability modes on the high regional frequencies of CWP extremes across multiple regions in the same winter (i.e., spatially compounding events) cannot be deduced **directly. For example, based on Metric 1, we find that the variability mode phase ENSO+ modulates regionally averaged CWP extreme frequencies for North America and Central Asia. However, a possibility could be that half of the winter seasons with ENSO+ leads to increased CWP extremes for North America only, while the other half affects Central Asia, thus not simultaneously. Examining the dependencies between counts of CWP extremes in different regions can provide preliminary information on spatially compounding events (Bevacqua et al., 2021) because regions connected by positive dependencies tend to experience CWP extremes at the same time. Thus, as a first step for investigating spatially compounding events, we analyse dependencies in Metric 1 computed for different regions, so as to provide preliminary information on groups of regions that may be affected by CWP extremes during the same winters.**

**Then, to examine spatially compounding events, we use two additional metrics. We employ these metrics to investigate the effects of variability modes on regional high frequencies of CWP extremes** across multiple regions in the same winter (Metric 2) and on the total population of the Northern Hemisphere exposed to CWP extremes in the same winter (Metric 3)."

**Comments:**

18. L328-331 this is motivation, should maybe go to introduction.

**Response:**
We agree with the reviewer that this sentence is misplaced. Instead of removing it, we decided to move and adapt this sentence when providing the results for Figure 4 at the beginning of subsection 3.2:

"**The presence of a given mode phase or combination of mode phases that has an effect on CWP extremes in multiple regions suggests the potential for spatially compounding events, which will be examined in subsection 3.3.**"

**Comments:**
19. L333: "Figure 5a shows Spearman correlations of regionally averaged CWP extreme frequencies (Metric 1) between all pairs of regions" You regionally average CWP extreme frequencies, but I'm thinking this could be slightly problematic. The regions are quite large, whereas these CWP extremes can be very local. What happens when you sum the CWP counts instead? Also, why not perform a spatial-dependency analysis on the original CWP data on high frequency, e.g. monthly?

**Response:**
Thanks for this comment. As with any dependency analysis, a choice on the spatial or temporal scale has to be taken. First, to avoid a potential misunderstanding, we note that we do not average precipitation and wind across the region but rather the counts of CWP extremes, thereby accounting for the local-scale dependency between precipitation and wind extremes. Generally, we agree that regional averaging of CWP extreme frequencies does not allow for analysis of dependency between local-scale CWP extremes, potentially occurring at the local scale, within different regions.

Here, with our choice of the considered spatial scale, by averaging CWP frequencies across a region within a winter, we aim to define an indicator of conditions that can generally be detrimental for the considered region within a winter. Then, we study spatial dependencies between such regional-scale detrimental conditions. Although we recognize that the considered spatial scale does not allow us to investigate CWP extremes at local scale, we note that our study complements the work carried out using the SREX regions adopted by the Intergovernmental Panel on Climate Change (IPCC). We clarified in the text this by adding:

L83: "We examine the influence of four variability modes on CWP extremes across 25 selected regions in the Northern Hemisphere defined in the SREX (see Fig. 1, Iturbide et al., 2020). **We chose these regions as they are standard reference in IPCC reports, as they encompass areas with relatively homogeneous climatic characteristics (Iturbide et al., 2020). While using these regions does not enable an explicit analysis of dependencies between local-scale CWP extremes and modes of variability, it allows for complementing IPCC assessments.**"

We also mention in the Discussion that the influence of variables on more local CWP extremes can be investigated in future research:

"In addition, the selected SREX regions may not reflect the natural spatial patterns of variation of CWP extremes**, potentially occurring at a more localized scale or span across multiple regions."**

Regarding the choice for the timescale, we already mentioned earlier that shorter timescales (e.g., weekly or monthly indices) could have allowed us to capture the variability of extremes more precisely. However, the choice of the seasonal scale in this study allows to account for 1) the integrated influence of some important modes like ENSO, which operates primarily on seasonal to annual timescales (e.g., Schmidt et al., 2001; Camberlin et al., 2001), but also 2) the interactions between variability modes occurring at seasonal timescales. Looking at shorter timescales would be a different study as it leads to different physical connections and mechanisms, and is therefore an interesting question to explore in future research.

Concerning the suggestion of the referee, which is to sum CWP counts over the regions instead of averaging, we note that this would not allow us to examine local CWP extremes. This is for the same reason as for averaging: when summing over a large region, the total count might be dominated by a high number of grid cells with moderate CWP extremes, potentially masking local areas experiencing extreme CWP events. Summing CWP counts instead of averaging them over regions would yield the same results up to a scaling factor, with the scaling factor being the number of grid cells used to compute the average. Thus, Spearman correlations between regions would not be affected, as the rank of the metric for the individual regions would not be modified. Averaging facilitates a more equitable comparison of CWP extremes across regions of different sizes, a distinction that would not be achievable using summed values.

**Comments:**
20. L374-376: "In particular, variability modes in isolation do not lead to significant effects on the population exposure compared to neutral conditions, indicating the importance of considering combinations of modes to distil the effects of modes of variability on the population affected." Where do you draw this conclusion from? To me it is unclear how this is related to fig 6a (which you reference the sentence before).

**Response:**
Thanks for this comment. Indeed, this sentence refers to Fig. 6b. We added a reference to the appropriate panel in the text.

**Comments:**
21. Fig 7b: but NAO is an index of SLP, so in this sense when you compare NAO- to NAO neutral you will of course find a difference in SLP. Here you go into discussion how a NAO-can physically lead to more CWP extremes as discussed in other literature. This should not be a result in my opinion, unless you have actually shown a physical mechanism in your results (e.g. convection anomalies, wind anomalies, latent heating anomalies,…). I think this last section of the results is mostly repetition from the previous sections and can be taken out. Instead focus on interpreting these physical mechanisms in a discussion section.

**Response:**
Thanks for this comment. First, we removed the element of discussion regarding NAO- and the links to the literature from subsection 3.4, as indicated by the reviewer. We also removed

another element of discussion regarding the influence of model biases on results, that we now discuss in the Discussion section.

Second, although we agree with the reviewer that Figure 7 and subsection 3.4 can present some repetitions with previous subsections (e.g., regarding the identification of regions affected by the combinations of modes, already done in Figure 3), we still think that linking the affected regions to SLP patterns under combined modes having the most significant effects on spatially compounding events according to Metric 2 and 3 is an interesting results. Note that not all modes are defined based on SLP.

We suggest removing some mentions of results that have been already detailed in Fig. 3, which shorten the subsection. We leave all the explanations regarding SLP patterns and how they are changed depending on the combinations of modes, the main point of this Figure.

**Comments:**
Discussion
22. It is important to mention you use a climate model in the first sentence already

**Response:**
Thanks for this comment. We now mention the use of a climate model in the first sentence.

**Comments:**
23. Some sentences are unclear, e.g. L 428: "Simulations show that extreme spatially compounding events with many regions under CWP extremes in the same winter are enabled by positive dependencies between CWP extremes across different regions"

**Response:**
Thanks for this comment. This sentence is not part of the Manuscript anymore. We also worked on reformulating other sentences to improve the readability.

**Comments:**
24. L339: "Our model evaluation against ERA5 reanalysis data indicates that the simulated anomalies in CWP extremes associated with modes of variability are well suited for the purpose of our analysis (Figs. S2-S5)". In my opinion there's some differences between the ERA5 and CESM figures; notably, ERA5 seems more pronounced. There are also regions where ERA5 does not agree with CESM: e.g. S2 shows that parts of North America have a negative ratio in ERA5 under NAO+ whereas this is positive in CESM, or S3 shows parts of North Africa have differences for ENSO+. I think it is important to highlight this, because that means that for some regions we can not make strong statements.

**Response:**
Thanks for this comment. We now highlight in the Discussion the fact that ERA5 does not agree with CESM for some modes and regions (ENSO+ and Northern Africa, Fig. S3), and that, consequently, conclusions should be made with care.

**Comments:**

25. Why didn't you include IOD if you mention this has an influence on CWP extremes? To me this comes back to the general motivation for this study; the choice for these exact modes need to be motivated clearly.

**Response:**

Thanks for this comment. This is already addressed in our response to the referee's second comment.